# Bacterial extracellular vesicles as recyclable nutrient reservoirs

Astrid Laimer-Digruber [1,2,10], Tanja V. Edelbacher [1,3,9,10], Masoumeh Alinaghi[1], Mia S. C. Yu [3], Dapi Menglin Chiang [3,4,5], Benedikt Kirchner[3,5], Susanne I. Wudy [6], Waltraud Tschulenk[7], Ingrid Walter[7], Stefan Kummer [8], Christina Ludwig [6], Jan Přibyl [2], Michael W. Pfaffl[3] & Monika Ehling-Schulz [1] ✉

Bacterial extracellular vesicles (EVs) are known to mediate intercellular communication, virulence, and immune modulation. Here we show that bacteria can utilise EVs also as recyclable nutrient reservoirs. Using *Bacillus cereus* as a model organism, we demonstrate that EVs exhibit distinct dynamics depending on growth conditions: EVs produced in complex nutrient-rich media undergo time-dependent degradation, while those produced in defined nutrient-limited conditions remain stable and accumulate. We observe similar EV degradation patterns in *Staphylococcus aureus*. Time-resolved multi-omics profiling reveals that EVs containing the lipid sphingomyelin undergo progressive degradation. Using pharmacological inhibition, knockout mutants, and enzymatic complementation, we show that this process is driven by secreted sphingomyelinase (SMase). This enzyme contributes to degradation of sphingomyelin-containing EVs, thereby releasing their biomolecular cargo which can be used as a nutrient source. Growth assays confirm that SMase-mediated EV degradation provides a growth advantage when nutrients become depleted, thus establishing EVs as dynamic nutrient reservoirs.

Adaption to environmental conditions is a prerequisite for the persistence and survival of bacterial populations. Multiple mechanisms and diverse lifecycle styles, including spore formation[1] and post-mortem nutrient acquisition[2], are employed by bacterial cells to adapt to specific niches and overcome challenges posed by adverse environmental conditions. Here, we report on a yet-to-explore strategy for bacterial adaption: extracellular vesicles (EVs). EVs, which are found in all three kingdoms of life[3], are lipid bilayer-enclosed spheres composed of the biomolecular building blocks lipids, proteins and nucleic

acids[4]. While lipids are the major structural component defining EV morphology, proteins and nucleic acids are attached to or contained within EVs[5]. Over the last decade, EVs have gained increasing prominence as vehicles for RNA therapeutics[6], as vaccine platforms[7] and as biomarkers for cancer[8] and infections[9–11].

Although bacterial EVs are thought to be involved in various biological processes, research has mainly focused on their role in pathogenicity[12,13]. For instance, we have recently demonstrated that EVs of the Gram-positive human pathogen *Bacillus cereus* are

[1]Institute of Microbiology, Centre of Pathobiology, Department of Biological Sciences and Pathobiology, University of Veterinary Medicine, Vienna, Austria. [2]CEITEC MU, Masaryk University, Brno, Czech Republic. [3]Division of Animal Physiology and Immunology, School of Life Sciences Weihenstephan, Technical University of Munich (TUM), Freising, Germany. [4]Department of Biomedicine, University of Basel, Basel, Switzerland. [5]Institute of Human Genetics, University Hospital, LMU Munich, Munich, Germany. [6]Bavarian Center for Biomolecular Mass Spectrometry (BayBioMS), TUM School of Life Sciences, Technical University of Munich (TUM), Freising, Germany. [7]Institute of Morphology, Centre of Pathobiology, Department of Biological Sciences and Pathobiology, University of Veterinary Medicine, Vienna, Austria. [8]VetCore Facility for Research, University of Veterinary Medicine, Vienna, Austria. [9]Present address: CDL Research, University Medical Center Utrecht, Utrecht, The Netherlands. [10]These authors contributed equally: Astrid Laimer-Digruber, Tanja V. Edelbacher. ✉e-mail: monika.ehling-schulz@vetmeduni.ac.at

biologically active and facilitate shuttling of tripartite toxins to host cells in a protected manner, where they elicit inflammatory responses[14]. As mediators of virulence, EVs can also participate in biofilm formation[15], contribute to antibiotic resistance[16], and inhibit the growth of other competing bacterial species[17]. Although progress has been made in understanding the functions of bacterial EVs, the role of the bacterial EV lipid bilayer remains poorly understood. For instance, sphingolipids are repeatedly reported as major constituents of bacterial EV lipid bilayers[18–21]. Yet, the majority of bacterial species, including *B. cereus*, is unable to synthesise sphingolipids de novo[22,23]. Nevertheless, they may be capable of scavenging sphingolipids from their environment[24]. Despite consensus on the presence of sphingolipids in bacterial EVs, their functional role within bacterial EV membranes remains to be elucidated.

Contrary to the prevailing concept of EVs accumulating over time in the extracellular space of bacteria, here we demonstrate that EVs derived from bacterial cultures grown in complex nutrient-rich LB-Miller (LB) medium or host factor-enriched RPMI (hfRPMI) medium undergo degradation. Notably, EVs from bacterial cultures grown under nutrient-limited conditions (MOD and RPMI medium) do not degrade but accumulate over time, indicating distinct functionalities of bacterial EVs tightly linked to environmental factors. We exclude inherent instability, biophysical features, or adjacent cells as the cause of the particular dynamics and behaviour of these EVs. Using a multi-omics approach and complementary functional studies, we demonstrate that EV degradation requires two key factors: sphingolipid-rich EV architecture and the presence of sphingomyelinase (SMase). Our work reveals that SMase-mediated EV-degradation is integral for bacterial adaption. In response to starvation, only SMase-targetable EVs consisting of sphingomyelins can serve as nutrient reservoirs. This physiologically relevant interaction between SMase and the EV lipid architecture could support bacterial adaption to distinct environments. Given the conservation of SMase and the structural similarity between bacterial and eukaryotic lipids, deciphering these enzyme-EV interactions will redefine the boundaries of EV biology.

## Results

### EV secretion dynamics in complex nutrient-rich and defined nutrient-scarce environments

To investigate how nutrient complexity affects EV secretion in Gram-positive bacteria, we compared *B. cereus* EV yields from complex, nutrient-rich (LB) and defined, nutrient-scarce (MOD) media over time. The emetic *B. cereus* reference strain F4810/72, which served as a model organism, was grown either in LB or in MOD. EVs were isolated from bacterial cultures after 0, 3, 6, 9, 12 and 15 h by differential centrifugation followed by ultracentrifugation, as outlined in Fig. 1a. The EVs isolated from *B. cereus* cultures were designated as $EVs_{LB}$ or $EVs_{MOD}$, respectively. Immediately after inoculating LB and MOD with *B. cereus*, as well as in sterile media controls, no EVs were detectable (Fig. 1b). However, a marked increase in *B. cereus* EV yields occurred during the initial 6 h of cultivation. Specifically, a 70-fold increase in EV yields was observed between 3 and 6 h in both LB and MOD, resulting in comparable EV yields of $EVs_{LB}$ and $EVs_{MOD}$ at 6 h. Unexpectedly, EV yields decreased significantly after 9 h in LB but remained stable in MOD (Fig. 1b). This decrease might either reflect a different stability of $EVs_{LB}$ and $EVs_{MOD}$ or might be linked to distinctive factors actively degrading $EVs_{LB}$.

### Biophysical properties of EVs or bacterial presence do not drive EV degradation

To test whether the observed differences in the dynamics of bacterial EV decrease relate to distinctive biophysical characteristics, we assessed biophysical properties of EVs using atomic force microscopy (AFM, Fig. 1c–f and Supplementary Fig. 1a–f). In addition, we employed two complementary transmission electron microscopy (TEM) approaches: (1) imaging of ultrathin sections prepared from resin-embedded *B. cereus* EVs that were subsequently stained (Supplementary Fig. 2a, b), and (2) the drop-on-grid method, in which *B. cereus* EVs were directly dropped onto TEM grids (Supplementary Fig. 2c, d). The TEM analyses confirmed that *B. cereus* EVs deriving from both media are lipid-bilayer bordered entities that are sphere-shaped. In the sterile media (LB, MOD) controls, no particle-like structures were detected by TEM (Supplementary Fig. 2e, f). Nanoparticle tracking analysis (NTA; Supplementary Fig. 2g) was used to assess the size of EVs. $EVs_{LB}$ (NTA: 195.3 nm) were found to be larger than $EVs_{MOD}$ (NTA: 142.3 nm), resulting in a size difference of 53.0 nm, which was validated by isolating EVs from three additional pathogenic *B. cereus* strains (Supplementary Fig. 2g). $EVs_{LB}$ and $EVs_{MOD}$ showed similar stability in AFM analysis, consistent with the observation that EVs remain intact when spiking them into sterile LB or MOD medium (Fig. 1g). These results highlight the inherent stability of both $EVs_{LB}$ and $EVs_{MOD}$, suggesting that other factors mediate the degradation of EVs.

To decipher the role of direct contact with adjacent bacteria in EV degradation, bacteria were cultivated for 7 h ($t_0$) in the respective media. Thereafter, bacterial cells were removed to produce cell-free conditioned media. The decrease in EV numbers in cell-free conditioned medium was determined after 4.5 h. As expected, no significant reduction of endogenous $EVs_{MOD}$ was observed. In contrast, endogenous $EVs_{LB}$ were found to be degraded in cell-free conditioned medium (Fig. 1h), indicating that bacterial contact is not required for EV degradation.

### EV degradation in host-factor-enriched RPMI parallels results from complex lab cultivation medium

Next, we asked whether EV degradation in LB and EV stability in MOD are specific phenomena in these particular growth environments or if they represent a more general feature of bacterial EVs associated with distinct environments. To answer this question, we compared the dynamics of EV production and degradation in two additional environments: host-factor-enriched RPMI medium and unconditioned RPMI medium (Supplementary Fig. 3a). To produce host-factor-enriched RPMI, Caco-2 cells were first differentiated in RPMI medium. Subsequently, the supernatant of the Caco-2 cells was collected by centrifugation, and host EVs were depleted by ultracentrifugation. This RPMI medium enriched with host factors and depleted of host EVs is designated as hfRPMI. The number of *B. cereus* EVs was monitored in bacterial cultures grown in parallel in hfRPMI and unconditioned RPMI. In concordance with the results from experiments in LB and MOD, a significant decrease in bacterial EVs was observed after 4.5 h of cultivation in hfRPMI, but no change in bacterial EV counts was observed in RPMI (Supplementary Fig. 3a, b). These findings imply that EV degradation and EV stability are not specific to LB and MOD but rather appear to be linked to the complexity of the environmental nutrient composition.

### EV degradation is not species-specific but a more common trait in bacteria

To test whether EV degradation is specific for *B. cereus* EVs, or whether it also occurs in other bacteria, *Staphylococcus aureus* Newman was cultured in different growth media (LB, hfRPMI, MOD, and RPMI) at 30 °C (Supplementary Fig. 3b). Subsequently, EV degradation was assessed by quantifying the number of EVs remaining in the conditioned media from each growth condition. In line with the results for *B. cereus* EVs, *S. aureus* Newman EVs were degraded in LB and hfRPMI, but not in MOD and RPMI (Supplementary Fig. 3b), suggesting that distinct but conserved factors influence EV degradation in bacteria.

### Compositional traits of EVs are associated with distinctive EV degradation dynamics

Still, it is unclear whether secreted bacterial factors released into the conditioned media or specific compositional features of EVs

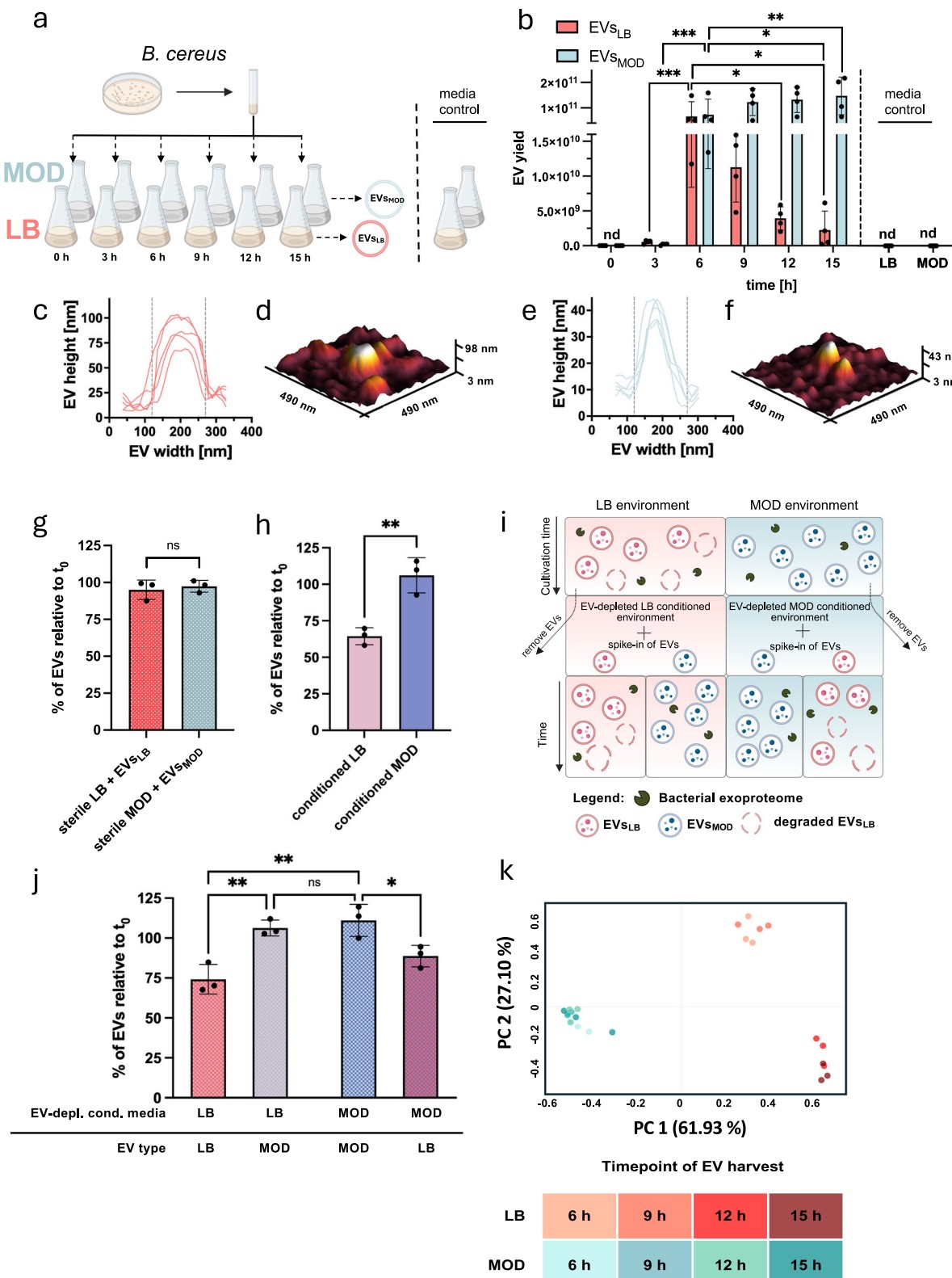

contribute to the degradation or stability of the EVs. To investigate this further, $EVs_{LB}$ or $EVs_{MOD}$ were each spiked into cell-free conditioned EV-depleted LB and MOD, and EV degradation was assessed (Fig. 1i, j). Notably, $EVs_{LB}$ were significantly prone to degradation in both environments, whereas $EVs_{MOD}$ remained stable (Fig. 1j). This suggests that $EVs_{LB}$ degradation is directly linked to the EV composition. In order to gain insights into the molecular components of EVs and their modulation over time in the two different environments, Fourier-transform

infrared spectroscopy (FTIR) was employed (Fig. 1k). FTIR is a high-resolution vibrational spectroscopic technique that provides detailed insight into the molecular composition of biological samples[25,26]. Based on stretching and bending vibrations of functional groups of the molecular constituents, encompassing lipids, proteins, polysaccharides and nucleic acids, global molecular fingerprints can be generated[14,27]. Chemometric analysis of spectral data recorded from $EVs_{LB}$ and $EVs_{MOD}$ revealed that the cultivation medium exerts the

**Fig. 1 | Distinct environmental conditions impact EV secretion and mediate EV degradation. a** To investigate EV secretion in nutrient-rich and nutrient-scarce cultivation environments, EVs of *B. cereus* F4810/72 were harvested 0, 3, 6, 9, 12 and 15 h post-inoculation—as depicted in the experimental scheme—and subjected to differential (ultra)centrifugation to obtain EVs. Media controls were prepared by subjecting sterile media to ultracentrifugation. **b** EVs were isolated from bacterial cultures via ultracentrifugation. The pelleted EVs were reconstituted in 30 µl PBS and EV yields were determined by nanoparticle tracking analysis (NTA) for each time point as indicated on the *x*-axis (*n* = 4 biological replicates). In addition, sterile media controls were subjected to the same EV isolation protocol described above and NTA was performed (*n* = 3 independent media preparations, nd: not detected). Atomic force microscopy (AFM) of EVs$_{LB}$ (**c, d**) and EVs$_{MOD}$ (**e, f**) reveal vesicular structures further visualised by line diagrams (**c, e**; *n* = 5 biological replicates). **g** To assess whether EVs degrade in sterile media, EVs$_{LB}$ or EVs$_{MOD}$ were spiked into the sterile media, and EV counts were determined after incubation for 4.5 h at 30 °C, 120 rpm (*n* = 3 independent experiments) significant EV degradation was measured in sterile culture media. **h** To determine whether bacterial cells are required for the degradation of EVs, bacteria were grown for 7 h. Subsequently, the bacterial cells were removed by means of centrifugation and sterile filtration, thereby obtaining cell-free, sterile media containing endogenous EVs. These media were designated as 'cell-free conditioned media' (*n* = 3 independent experiments). EV counts were determined after incubation for 4.5 h at 30 °C, 120 rpm (*n* = 3 independent experiments) directly in the conditioned media. A significant reduction in EV numbers was observed in cell-free conditioned LB but not in cell-free conditioned MOD. **i, j** To test whether EVs degrade due to the extracellular milieu or due to inherent EV characteristics, EVs$_{LB}$ and EVs$_{MOD}$ were purified and spiked to LB or MOD EV-depleted conditioned media, resulting in four conditions (LB conditioned media with EVs$_{LB}$; LB conditioned media with EVs$_{MOD}$, MOD conditioned media with EVs$_{MOD}$, MOD conditioned media with EVs$_{LB}$). **j** EVs were quantified at the start of the experiment (*t$_0$*) and after 4.5 h (*n* = 3 biological replicates). **k** Principal component analysis (PCA) of FTIR spectra was performed to decipher the effect of growth media on the composition of EVs. For this analysis, the average spectra of EVs from three biological replicates, each with four technical replicates, were calculated. Statistical significance was assessed using two-way ANOVA with Bonferroni's multiple comparison test (**b**), two-tailed *t*-test (**g, h**) and one-way ANOVA with Bonferroni's multiple comparison test (**j**). Data presented reflect mean ± SD. \**p* < 0.05, \*\**p* < 0.01, \*\*\**p* < 0.001, \*\*\*\**p* < 0.0001. **a** Created in BioRender. Ehling-Schulz, M. (2026) https://BioRender.com/gxymec6. **i** Created in BioRender. Ehling-Schulz, M. (2026) https://BioRender.com/w57bv8k. EVs$_{LB}$ EVs isolated from *B. cereus* cultures grown in LB, EVs$_{MOD}$ EVs isolated from *B. cereus* cultures grown in MOD. Source data are provided as Source Data file.

highest impact on the overall composition of the EVs. Principal component analysis (PCA) showed that PC1, which is associated with the growth medium, accounts for 61.9% of the variance. It is noteworthy that PC2 (explaining 27.1% of the variance) further differentiates EVs$_{LB}$ based on the sampling time point into two clusters. One cluster was constituted by EVs$_{LB}$ from 6 h and 9 h, while the EVs$_{LB}$ from 12 h and 15 h formed a separate, distinctive cluster. In contrast, EVs$_{MOD}$ from all four sampling points clustered together, indicating that the composition of EVs$_{MOD}$ does not undergo significant alteration over time (Fig. 1k). These results emphasise that the compositional traits of EVs$_{LB}$ and EVs$_{MOD}$ follow distinctive dynamics, prompting further analysis through untargeted proteomics and lipidomics.

**Proteome remodelling reflects EV degradation in LB**

To investigate the potential impact of EV degradation on the *B. cereus* EV proteome, we performed an untargeted LC-MS/MS proteomic analysis of EVs$_{LB}$ and EVs$_{MOD}$ harvested from bacterial cultures after 9 h and 15 h. A list of identified proteins is provided in Supplementary Data 1. The majority of proteins in EVs$_{LB}$ are uniquely present at 9 h, while most EVs$_{MOD}$ proteins are present at both time points (Fig. 2a, b). PCA was performed on label-free quantification (LFQ) intensity data for unsupervised multivariate statistical analysis. Samples were primarily clustered by medium along PC1 (PC1, 60.5%; Fig. 2c), with EVs$_{LB}$ also diverging into clusters based on cultivation time along PC2 (PC2, 23.4%; Fig. 2c). In contrast, EVs$_{MOD}$ samples from both cultivation time points clustered together. This is in line with the results from FTIR spectroscopic analysis (Fig. 1k), which support the notion that the EV$_{LB}$-compositional profile undergoes dynamic changes. The high number of differentially regulated proteins (DEPs) observed between the EVs$_{LB}$ 9 h and EVs$_{LB}$ 15 h proteomes pinpoints the impact of EV degradation on the EV proteome (Supplementary Fig. 4a). On the contrary, EVs$_{MOD}$ proteomes show only minor changes between 9 and 15 h (Supplementary Fig. 4b). The list of proteins included in the differential abundance analysis is provided in Supplementary Data 2. The DEPs in *B. cereus* EVs$_{LB}$, compared to *B. cereus* EVs$_{MOD}$, determined at two different timepoints (9 and 15 h; Supplementary Fig. 4c, d), were subjected to Gene Ontology (GO) enrichment analysis. This resulted in the identification of enriched pathways for each comparison. Because the GO hierarchy contains many redundant or overlapping terms, a redundancy-reduction step was applied to obtain a non-redundant set of pathways. Redundant GO terms were identified using a semantic similarity-based clustering implemented in the Python workflow with GOATOOLS[28]. The enriched pathways, cleared of duplicate and redundant terms, were visualised using a Sankey Diagram (Fig. 2d). This analysis revealed an overlap in enriched pathways at 9 and 15 h in EVs$_{MOD}$, but not in EVs$_{LB}$. In summary, these results indicate active proteome remodelling in EV$_{LB}$ but not in EV$_{MOD}$.

**Growth environment impacts EV lipid architecture deciphered by lipidomics**

Since the lipid bilayer directly interacts with the extravesicular milieu, and EV degradation has been found to depend on culture conditions, we hypothesised that the lipid bilayer may play a distinctive role during this process. Thus, we next carried out untargeted lipidomic analysis of EVs.

This analysis revealed a distinctive lipid bilayer composition of EVs$_{LB}$ and EVs$_{MOD}$. A total of 544 lipid species for EVs$_{LB}$ and 311 lipid species for EVs$_{MOD}$ were identified. The list of identified lipids is presented in Supplementary Data 3. Among all lipid species, a total of 295 lipid species were shared between EVs$_{LB}$ and EVs$_{MOD}$ (Fig. 3a). Lipid species were further classified based on the LIPID MAPS® Lipid Classification System into four lipid classes: fatty acyls (FAs), glycerolipids (GLs), glycerophospholipids (GPs) and sphingolipids (SPs). Analysis of sterols, prenols, saccharolipids and polyketides was omitted, since less than two lipid species were detected. The lipid composition of EVs was characterised through the quantification of the relative abundances of individual lipid species. This analysis revealed a predominance of sphingolipids in the EVs$_{LB}$ lipidome, comprising 78.36% of the total lipid composition. This contrasts with the EVs$_{MOD}$ lipidome, which contains 0.08% sphingolipids (Fig. 3b and Table 1). Specifically, sphingomyelins and glycosphingolipids were only detected in EVs$_{LB}$ but not in EVs$_{MOD}$. To further investigate the sphingomyelin content in *B. cereus* EVs, a quantitative sphingomyelin assay was performed, revealing a dose-dependent increase in sphingomyelins in EVs$_{LB}$, whereas EVs$_{MOD}$ did not contain any detectable sphingomyelins (Supplementary Fig. 5a). To rule out contaminations from potentially pelletable particles in LB, we quantified sphingomyelin in sterile LB and sterile, ultracentrifuged LB (LB*) (Supplementary Fig. 5b). Both media showed similar sphingomyelin levels, indicating the absence of sphingomyelin-positive pelletable particles (Supplementary Fig. 5b).

The near absence of sphingolipids in EVs$_{MOD}$ resulted in major compositional shifts in the other lipid classes (Fig. 3b), as 73.44% of the EVs$_{MOD}$ lipidome consists of glycerolipids. The observed differences in the lipid compositions of EVs$_{LB}$ and EVs$_{MOD}$ were confirmed by chemometric analysis of FTIR data (Fig. 3c and Supplementary Fig. 6a). Subtraction spectra analysis revealed significant differences between

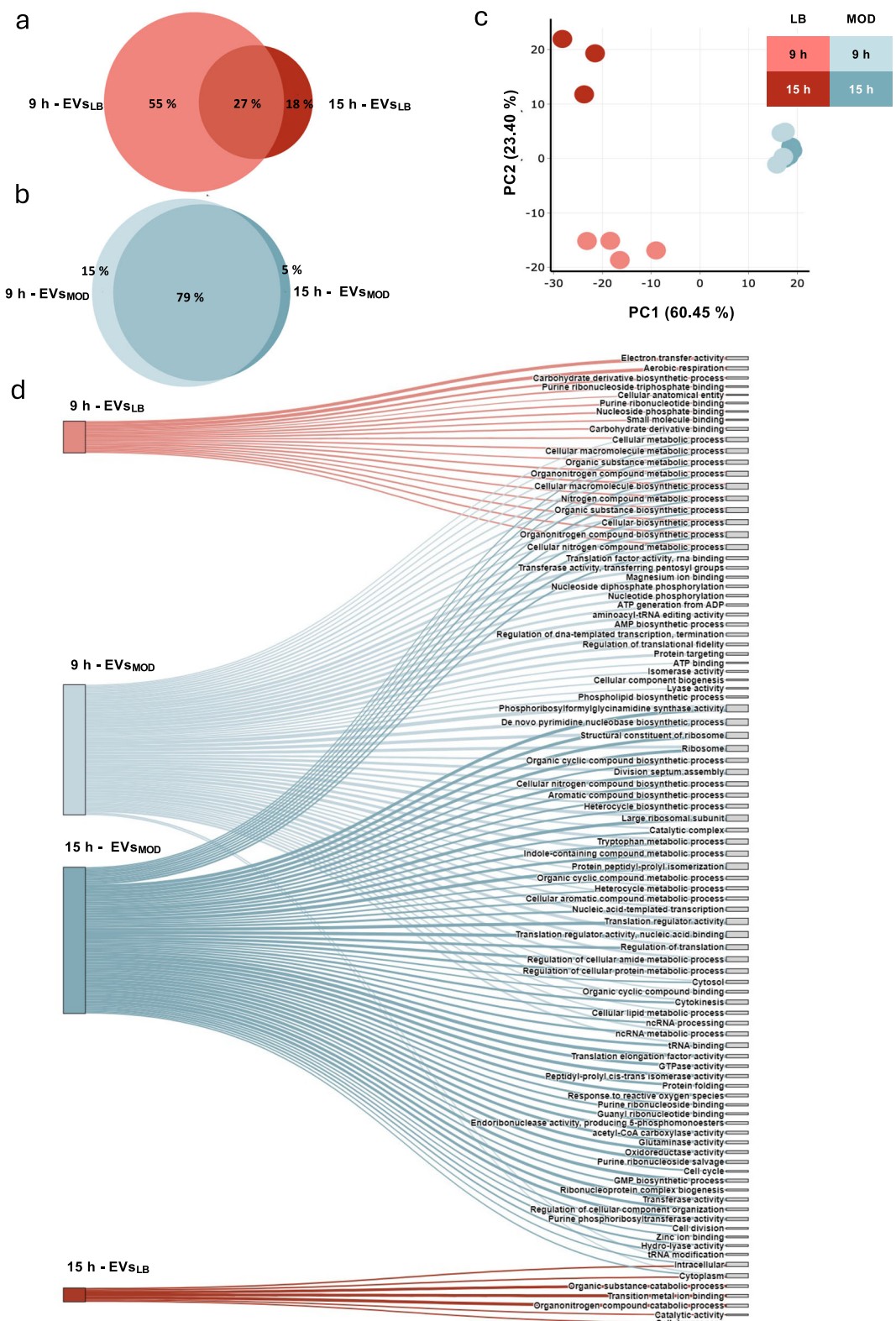

**Fig. 2 | Changes in the EV proteome give insights into EV degradation.** The cargo of *B. cereus* EVs was characterised by untargeted LC/MS-MS analysis. The Euler diagram shows the relative overlap of identified proteins in 9 and 15 h EVs$_{LB}$ (**a**), and 9 and 15 h EVs$_{MOD}$ (**b**). **c** EV proteomes (EVs$_{LB}$ 9 and 15 h; EVs$_{MOD}$ 9 and 15 h) were further compared by multivariate analysis (PCA) based on LFQ intensities, showing distinct clustering based on the cultivation conditions, time and media. **d** Differentially regulated proteins (DEPs) from the comparison 9 h EVs$_{LB}$ vs. 9 h

EVs$_{MOD}$ and 15 h EVs$_{LB}$ vs. 15 h EVs$_{MOD}$ (Supplementary Fig. 4) were used for GO-term mapping. Redundancy removed; enriched pathways (one-sided hypergeometric test, FDR-adjusted *p* value < 0.05) were visualised using a Sankey diagram. Edge widths correspond to fold-enrichment values. The nodes and edges are coloured based on the experimental conditions. **a–d** For proteomic analysis, three biological replicates were sampled for EVs$_{LB}$ 15 h, four biological replicates were used for all other samples (EVs$_{LB}$ 9 h, EVs$_{MOD}$ 9 h and EVs$_{MOD}$ 15 h).

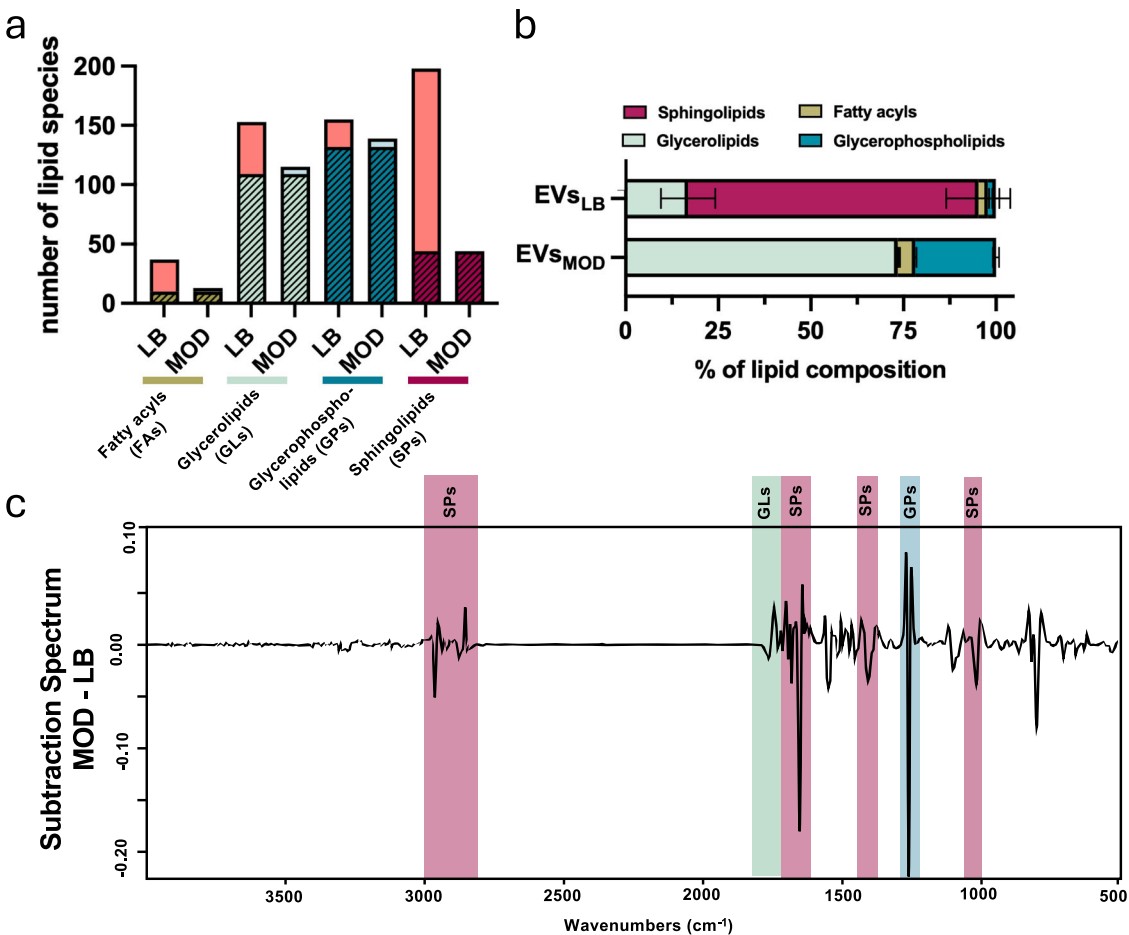

**Fig. 3 | Culture media influence the architecture of EV lipidome.** To collect information about EV building blocks, the EV lipid composition was investigated using an untargeted lipidomics approach and FTIR. **a** Untargeted lipidomic analysis of EVs revealed a wide distribution of lipid species. The number of lipid species of the respective lipid class (fatty acyls (FA), glycerolipids (GL), glycerophospholipids (GP), sphingolipids (SP)) was quantified. The overlap between $EVs_{LB}$ and $EVs_{MOD}$ is indicated by hatching ($n = 3$ biological replicates). **b** The relative lipid amounts of $EVs_{LB}$ and $EVs_{MOD}$ were calculated, showing major shifts in lipid compositions. Data presented reflect mean ± SD of three biological replicates. **c** The subtraction spectrum ($EVs_{MOD}$-$EVs_{LB}$) of FTIR 2nd derivative spectra was used to compare the spectroscopic profile of EVs, highlighting the differences in the regions accounting for sphingolipids (berry), glycerolipids (sage) and glycerophospholipids (petrol). For this analysis, the average spectra of EVs from three biological replicates, each with four technical replicates, were calculated. Source data are provided as Source Data file.

$EVs_{LB}$ and $EVs_{MOD}$ in regions specific to sphingolipids and glycerolipids (Fig. 3c). These findings were further corroborated by the analysis of the loading plot that separates the samples based on media along PC1 (Supplementary Fig. 6b). In conclusion, $EVs_{LB}$ and $EVs_{MOD}$ exhibit substantial differences in their lipid architecture, which may be associated with their distinct susceptibility to degradation processes.

**Sphingolipid composition dictates susceptibility to SMase-driven EV degradation**

Due to the elevated levels of sphingolipids, in particular sphingomyelins, found in $EVs_{LB}$, we hypothesised that a specific sphingomyelin-degrading enzyme secreted by *B. cereus* might be responsible for EV degradation. Given that *B. cereus* is known to secrete a sphingomyelinase (SMase, sphingomyelin phosphodiesterase; EC 3.1.4.12; see also Supplementary Fig. 7a)[29,30], which hydrolyses sphingomyelin to phosphocholine and ceramide, we investigated whether *B. cereus* SMase could be a possible driver of EV degradation. To achieve this, we employed four complementary approaches. First, to test whether *B. cereus* SMase can degrade its sphingolipid-rich $EVs_{LB}$, sterile LB was spiked with $EVs_{LB}$ and treated with commercially available *B. cereus* SMase (Fig. 4a, b). We observed a significant degradation of

$EVs_{LB}$ in the presence of SMase, indicating that *B. cereus* SMase can indeed degrade EVs (Fig. 4b). Next, we investigated whether endogenously secreted SMase could be inhibited, thereby stopping EV degradation (Fig. 4c, d). To inhibit SMase activity in cell-free conditioned LB media, we used the SMase inhibitor Imipramine. Indeed, adding Imipramine to cell-free conditioned LB containing both endogenous EVs and endogenous SMase led to significantly reduced EV degradation (Fig. 4d). Except for Aprotinin, the protease inhibitors tested in parallel did not inhibit this temperature-dependent, enzyme-driven process as effectively as the SMase inhibitor Imipramine (Supplementary Fig. 7b–d). Next, to assess whether protease inhibitors affect *B. cereus* SMase activity, we measured SMase activity with a fluorometric enzyme activity assay in the presence of Imipramine, Aprotinin, EDTA and PMSF, respectively (Supplementary Fig. 7e). Imipramine strongly inhibited SMase activity, confirming its direct inhibition of *B. cereus* SMase. Furthermore, Aprotinin inhibited SMase activity to a lesser but significant extent, while PMSF did not show a significant effect in the SMase activity assay. Aprotinin and PMSF are both serine protease inhibitors; however, their mechanisms of action differ, which may account for their divergent effects on SMase activity and SMase-mediated EV degradation. Consistent with the weak effect on EV degradation (Supplementary Fig. 7d), EDTA showed only weak

**Table 1 | Distribution of lipid main and sub-class species in percent among the total lipidome**

| Super class | Main class/Sub class | % of total lipidome in | |
|---|---|---|---|
| | | EVs$_{LB}$ | EVs$_{MOD}$ |
| Fatty acyls | Saturated | 2.07 ± 0.15 | 2.51 ± 0.21 |
| | Unsaturated | 0.43 ± 0.23 | 2.14 ± 0.13 |
| | Fahfa | 0.05 ± 0.00 | ND[a] |
| Glycerolipids | Diradylglycerols | 13.99 ± 7.68 | 69.41 ± 1.45 |
| | Glycosyldiradylglycerols | 0.28 ± 0.10 | 3.53 ± 0.25 |
| | Triradylglycerols | 2.58 ± 0.48 | 0.50 ± 0.77 |
| Glycerophospholipids | Cardiolipins | 0.47 ± 0.53 | 0.72 ± 0.23 |
| | Glycerophosphates | ND[a] | 0.05 ± 0.03 |
| | Glycerophosphocholines | 0.02 ± 0.00 | ND[a] |
| | Glycerophosphoethanolamines | 0.62 ± 0.08 | 4.61 ± 0.64 |
| | Glycerophosphoglycerols | 0.59 ± 0.42 | 16.38 ± 0.17 |
| | Glycerophosphoglycerophosphoglycerols | 0.47 ± 0.06 | ND[a] |
| Sphingolipids | Ceramides | 71.26 ± 7.83 | 0.08 ± 0.01 |
| | Glycosphingolipids | 5.93 ± 0.75 | ND[a] |
| | Sphingomyelins | 1.17 ± 0.03 | ND[a] |

[a]ND (not detected) indicates that the % abundance was <0.02 and was therefore omitted for this analysis.

inhibition in the SMase activity assay (Supplementary Fig. 7e). Given that SMase activity is dependent on divalent cations such as $Co^{2+}$ and $Mg^{2+}$[31], this weak inhibitory effect is likely to be associated with the chelation of divalent cations by EDTA[32].

To confirm the specificity of Imipramine under our experimental conditions, we spiked sterile LB media with EVs$_{LB}$ and SMase. The addition of Imipramine blocked EV degradation, demonstrating that Imipramine specifically inhibits SMase activity (Supplementary Fig. 7f). Finally, we used an isogenic SMase knock-out mutant (Δ$sph$) of *B. cereus* F4810/72 to produce cell-free conditioned LB medium lacking endogenous SMase. Conditioned LB medium created from the mutant Δ$sph$ and its parental wild type (WT) was depleted of EVs and subsequently spiked with EVs$_{LB}$ originating from the WT (Fig. 4e, f). The lack of SMase in the bacterial extracellular environment significantly reduced EV degradation (Fig. 4f). In summary, the results from these three complementary approaches foster our hypothesis that SMase plays a pivotal role in EV degradation.

To rule out the possibility of differential protease activity being the underlying cause of this effect, zymography was conducted using gelatin and casein substrates. This analysis revealed no significant differences in the proteolytic activity between the WT and the Δ$sph$ exoproteomes (Supplementary Fig. 7g). To confirm the pivotal role of SMase in EV degradation, we tested whether the lack of SMase would also impede EV degradation in continuously grown cultures (Fig. 4e, g). The WT and the Δ$sph$ mutant were cultivated in LB. The EV yields were assessed before EV degradation was observed in LB after 7 h ($t_0$) and then compared to EV yields after 10 h ($t_1$) and 13 h ($t_2$). As expected, EV yields decreased in WT bacterial cultures after prolonged cultivation but remained stable in Δ$sph$ bacterial cultures, indicating that the observed EV decline can be attributed to the presence of SMase in the bacterial exoproteome (Fig. 4e, g). To exclude the possibility that co-purified factors may affect EV degradation, EVs$_{LB}$ were further purified by density gradient ultracentrifugation (EVs$_{LB-DG}$). In addition, to rule out the possibility that LB-derived particles interfere with the EV degradation experiments, EVs from *B. cereus* grown in ultracentrifuged LB (LB*) were harvested by ultracentrifugation (EVs$_{LB*}$) and further purified by density gradient ultracentrifugation (EVs$_{LB*-DG}$). Subsequently, EVs$_{LB-DG}$ and EVs$_{LB*-DG}$ were used in spike-in experiments. Both EVs$_{LB-DG}$ and EVs$_{LB*-DG}$, were spiked into sterile LB, sterile LB supplemented with external SMase, or sterile LB containing both external SMase and Imipramine, and the results were compared to those from

spike-in experiments with EVs$_{LB}$ and EVs$_{LB*}$ (Supplementary Fig. 8a). Irrespective of the method used for *B. cereus* EV preparation, EV degradation was observed exclusively in the presence of SMase and was inhibited by Imipramine. No significant differences were observed in the extent of SMase-mediated EV degradation between EVs$_{LB}$, EVs$_{LB*}$, EVs$_{LB-DG}$, and EVs$_{LB*-DG}$ (Supplementary Fig. 8a). Furthermore, EVs purified by density gradient centrifugation tested positive for sphingomyelins (Supplementary Fig. 8b). Similar amounts of sphingomyelin were observed in EVs$_{LB-DG}$, EVs$_{LB*-DG}$, EVs$_{LB}$, and EVs$_{LB*}$. These findings demonstrate that density gradient-purified *B. cereus* EVs have the same sphingomyelin-positive lipid architecture and SMase susceptibility as crudely purified *B. cereus* EVs (Supplementary Fig. 8a, b). Consequently, these results reinforce the observation presented in Fig. 1b and demonstrate that the degradation of EVs$_{LB}$ requires two factors: a sphingolipid-rich EV architecture and the presence of SMase within the extravesicular milieu.

**Stress signatures distinguish EVs$_{MOD}$ from EVs$_{LB}$**

To elucidate the physiological implications of EV degradation, bacterial growth behaviour was profiled by calculating the maximum growth rate (Fig. 5a and Supplementary Fig. 9a) and the time to reach stationary phase (Fig. 5b and Supplementary Fig. 9a). Both parameters indicated that the nutrients in LB are consumed at a faster rate, as growth in LB was characterised by a significantly higher growth rate and a substantially shorter time to reach the stationary phase (Fig. 5a, b and Supplementary Fig. 9a). The response to these divergent environments manifested in different growth capacities and distinct compositions of the EV-cargo (Fig. 2d and Supplementary Fig. 4a–d). Therefore, we examined the proteomics dataset generated in this study from *B. cereus* EV (Supplementary Data 1; EVs$_{LB}$ 9 h, EVs$_{LB}$ 15 h, EVs$_{MOD}$ 9 h, EVs$_{MOD}$ 15 h) for proteins associated with environmental adaptation (Fig. 2). To obtain a comprehensive profile of the stress response proteome, the GO annotations for biological processes were first retrieved for all identified *B. cereus* EV proteins. Subsequently, the proteins associated with the parent GO term 'response to stimuli' (GO:0050896) and its child terms (descendant terms in the GO hierarchy structure) were selected. The intensities of the selected proteins were visualised in the heat-map depicted in Fig. 5c. This analysis revealed a total of 25 proteins that diverged into two distinct groups of stress-associated proteins (Fig. 5c). Notably, these two groups of proteins mirror the two particular growth conditions. The first group,

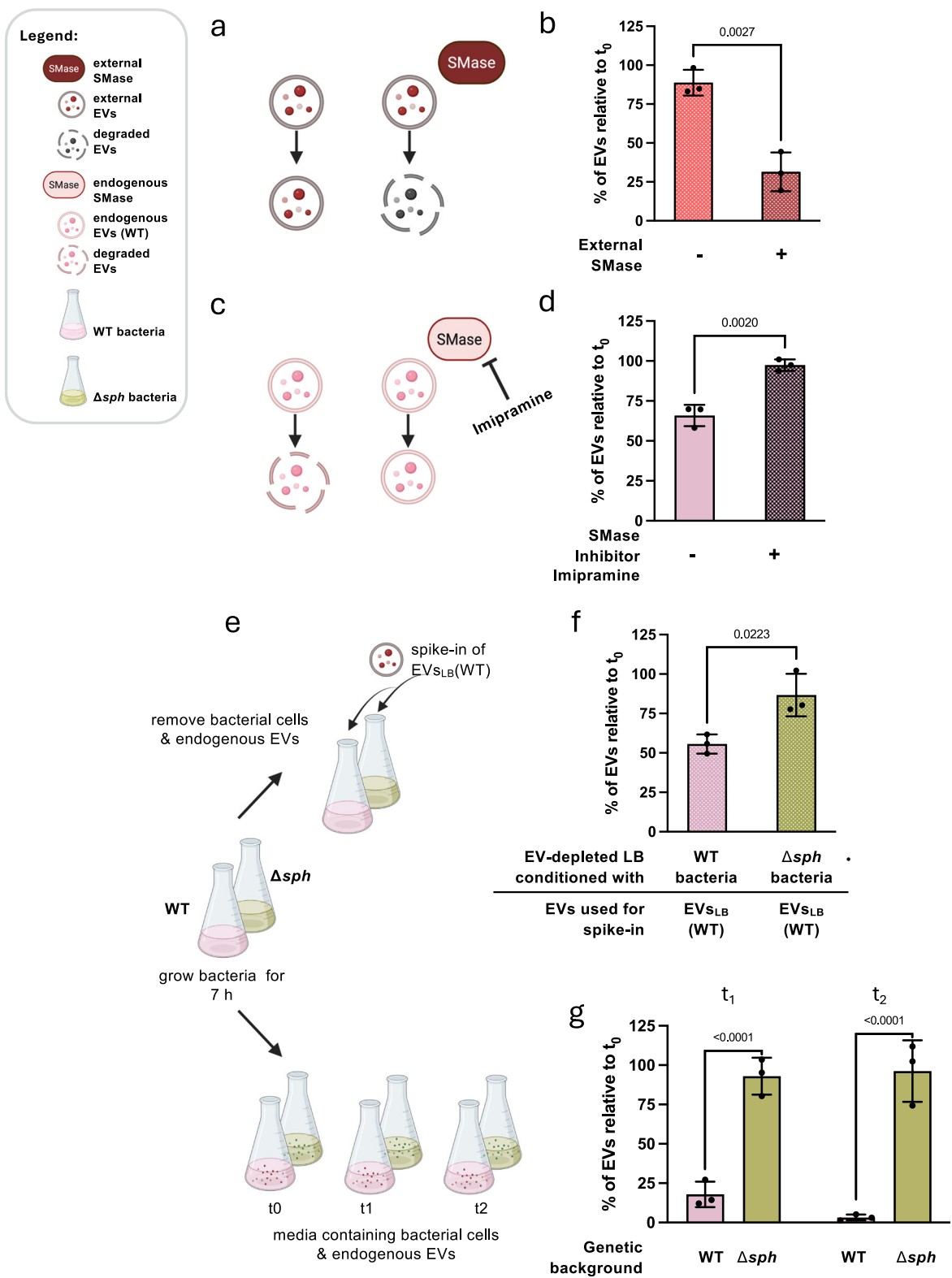

exclusively enriched in $EVs_{MOD}$, contained proteins associated with translational and oxidative stress, such as the DEAD-box ATP-dependent RNA helicase CshA and CshB, as well as the DNA repair protein RecN, and the superoxide dismutases SodA2 and SodC. The second group, exclusively enriched in $EVs_{LB}$, comprised proteins associated with nutritional stress (DNA protection during starvation protein 1 (Dps1) and 2 (Dps2), putative carbon starvation protein A). Generally, translational and oxidative stress are tightly linked, and the enrichment of these proteins in $EVs_{MOD}$ implies that MOD-cultivated bacteria are exposed to translational and oxidative stress. While those translational and oxidative stress proteins are barely abundant in $EVs_{LB}$, the exclusive enrichment of Dps1, Dps2 and the putative carbon starvation protein A in $EVs_{LB}$ suggests that bacteria cultivated in LB experience nutritional stress[33–35].

**Fig. 4 | EV degradation is SMase-mediated.** Four independent approaches were employed to assess the impact of SMase on EV degradation. **a, b** The susceptibility of $EVs_{LB}$ to exogenously added SMase was tested by adding commercially available *B. cereus* SMase to sterile LB with spike-in of $EVs_{LB}$. **c, d** The effect on EV degradation by inhibiting SMase activity with a commercially available SMase inhibitor was tested by applying Imipramine to cell-free conditioned LB media. **e, f** The effect of endogenously produced SMase on EV degradation was tested by spike-in of $EVs_{LB}$ originating from the WT to EV-depleted cell-free conditioned LB media stemming from either WT bacteria or the $\Delta sph$ isogenic knock-out mutant, lacking the gene encoding for SMase. **b, d, f** EVs were quantified 4.5 h after the start of the experiment and normalised to $t_0$ values. **e, g** To test whether the lack of SMase prevents EV degradation, bacterial cultures were grown for 7 h ($t_0$), and EV yields were measured in the continuously growing bacterial culture after 10 h ($t_1$) and 13 h ($t_2$; $n = 3$). Statistical significance was determined using an unpaired two-tailed $t$-test (**b, d, f**) or a two-way ANOVA with Bonferroni's multiple comparison test (**g**). Data presented reflect mean ± SD of three biological replicates. *$p < 0.05$, **$p < 0.01$, ***$p < 0.001$, ****$p < 0.0001$. **a, c** Created in BioRender. Ehling-Schulz, M. (2026) https://BioRender.com/9hlqh9n. **e** Created in BioRender. Ehling-Schulz, M. (2026) https://BioRender.com/eaoy2xl. Source data are provided as Source Data file.

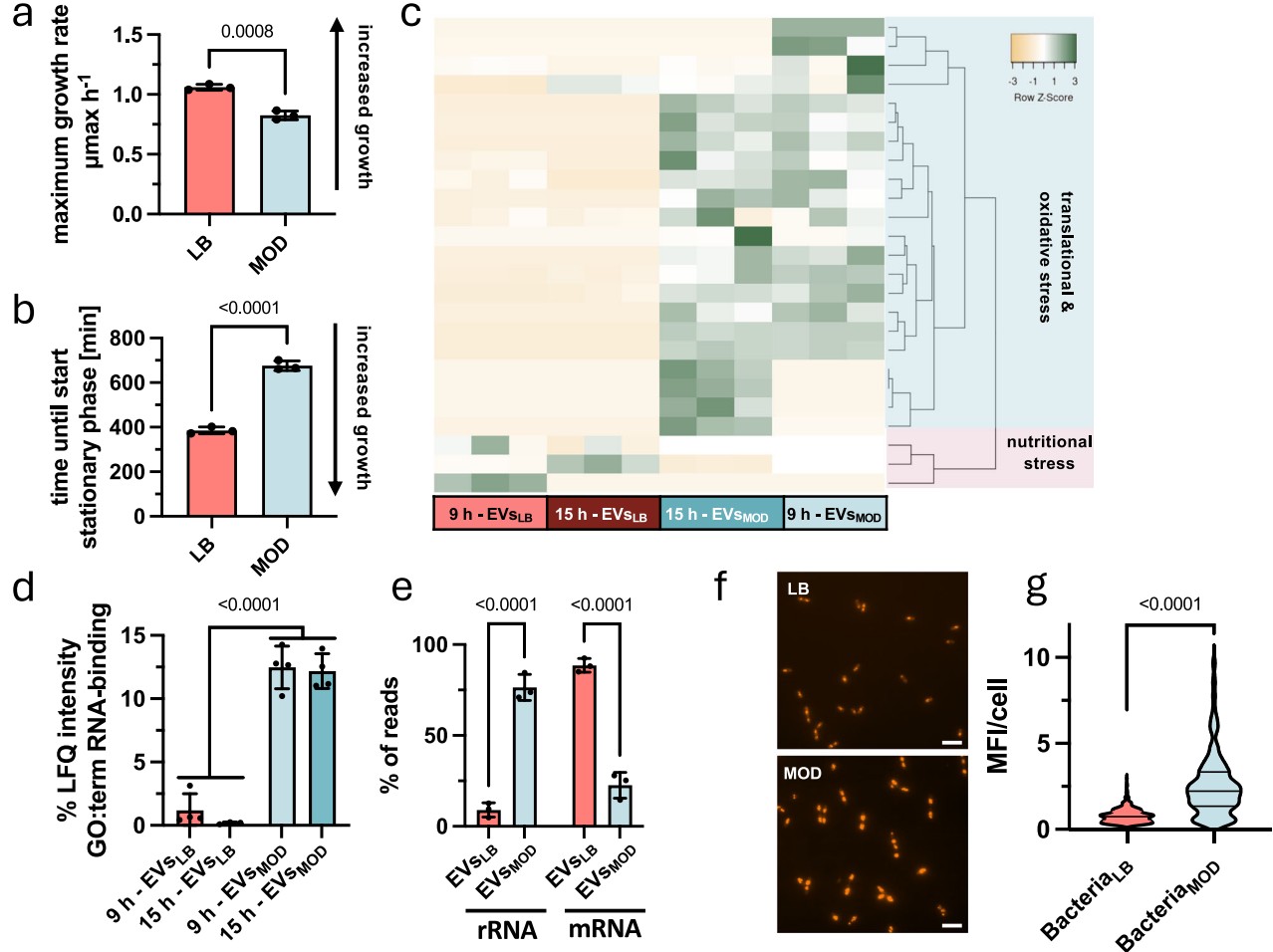

**Fig. 5 | Specific EV-cargo mirrors the metabolic state of bacterial cells.** Bacteria were cultivated in LB and MOD and the maximum growth rate (**a**) as well as the time until stationary phase (**b**) was calculated by formulas based on the Gompertz function ($n = 3$). **c** To gain insight into adaptive changes of bacterial cells, the EV-proteome was investigated for stress-associated proteins, by filtering for proteins with child-terms of the 'response to' GO-term. The heat map displays the abundance of 25 stress-associated proteins in $EVs_{LB}$ and $EVs_{MOD}$ harvested after 9 and 15 h. **d** Analysis of proteins associated with RNA-binding in the particular proteome shows a relative increase in $EVs_{MOD}$ ($n = 3$). **e** Analysis of the EV-transcriptome without rRNA-depletion revealed unique loading of RNA-types in $EVs_{LB}$ and $EVs_{MOD}$, primarily characterised by the increase of rRNA in $EVs_{MOD}$ ($n = 3$). **f, g** To detect oxidative stress and the resulting reactive oxygen species (ROS), CellROX™ staining was performed on 15 h LB- or MOD- grown bacteria. The production of ROS is illustrated in representative images (**f**) and additional images of biological replicates are shown in the Supplementary Fig. 9. **g** Mean fluorescence intensities were calculated using ImageJ ($n = 3$). The fluorescence intensities of 512 (LB-cultivated) or 483 (MOD-cultivated) bacterial cells were analysed and plotted. Statistical significance was determined by unpaired two-tailed $t$-test (**a, b, g**) or two-way ANOVA with Bonferroni's multiple comparison test (**d, e**). **a, b, d, e** Data presented reflect mean ± SD. *$p < 0.05$, **$p < 0.01$, ***$p < 0.001$, ****$p < 0.0001$. Source data are provided as Source Data file. Scale bar indicates 10 μm.

Translational stress in MOD-grown bacteria was also supported by the increase in the mass spectrometric intensity of proteins associated with RNA binding in $EVs_{MOD}$ (Fig. 5d). To validate this finding, we performed a transcriptomic profiling of 9 h EVs (Fig. 5e and Supplementary Data 5). This analysis revealed an increased abundance of rRNA in $EVs_{MOD}$ (Fig. 5e), corroborating the proteomic data. Given that the semantic GO-term-based profiling of stress-associated proteins indicated the presence of both translational and oxidative stress in $EVs_{MOD}$ (Fig. 5c), we further examined oxidative stress levels in bacteria (Fig. 5f, g and Supplementary Fig. 9b, c). Bacteria were grown for 15 h in LB and MOD, and reactive oxygen species (ROS) in bacterial cells were determined using CellROX™ Staining and fluorescence microscopy

(Fig. 5f and Supplementary Fig. 9b, c). As expected, significantly higher signals were measured for MOD-cultured bacteria than for LB-cultured bacteria (Fig. 5g). These results suggest that MOD-cultured bacteria experience oxidative and translational stress, while LB-cultured bacteria are facing nutritional stress, which is reflected in the proteomes of the respective EVs. Taken together, the specific EV lipid architecture combined with the particular EV cargo may indicate divergent functions of the respective EVs.

### Bacterial growth benefits from endogenous SMase-degradable EVs as nutrient reservoirs

The concurrent presence of three starvation proteins (Fig. 5c and Supplementary Data 4) and the accelerated bacterial growth imply that bacteria grown in complex, growth-promoting environments, such as LB (Fig. 5a, b), deplete nutrient resources more efficiently and faster than bacteria grown in nutrient-scarce, mineral-based media such as MOD.

Since EV degradation only occurred in LB cultures but not in MOD cultures, we hypothesised that LB-cultivated bacteria could utilise the $EVs_{LB}$ components to boost their growth. To test this hypothesis, we exposed bacteria that had been pre-cultivated for 7 h in LB or MOD ($t_0$) to either cell-free EV-depleted conditioned media or cell-free conditioned media containing endogenous bacterial EVs (Fig. 6a, b). The $OD_{600}$ was measured after 5 h ($t_1$) and 11 h ($t_2$) of prolonged growth to calculate the relative growth rate (Fig. 6a–c). The relative growth rate $t_1$–$t_2$ was significantly higher in LB media containing endogenous bacterial EVs compared to EV-depleted LB (Fig. 6c). In contrast, in MOD media, the presence of EVs did not significantly impact the bacterial growth rate (Fig. 6c). These findings emphasise a context-dependent role for sphingomyelin-rich EVs in promoting bacterial proliferation.

To further support the notion that $EVs_{LB}$ can be utilised as a nutrient resource, we leveraged the auxotrophy of *B. cereus* F4810/72 for valine[36]. Since the valine concentration in complex, nutrient-rich media cannot be controlled, we prepared MOD medium containing only minimal amounts of valine (0.7 mM), corresponding to 10% of the valine concentration in the standard MOD medium, designated $MOD_{Val*}$. $MOD_{Val*}$ was inoculated with SMase-producing WT bacteria and spiked with either $EVs_{LB}$ or $EVs_{MOD}$ (Fig. 6d, e). This approach revealed that only $EVs_{LB}$ can counterbalance the valine deficiency in $MOD_{Val*}$, emphasising that bacteria can utilise $EVs_{LB}$ as recyclable nutrient resources, whereas $EVs_{MOD}$ cannot. To provide further evidence that the interplay of SMase and $EVs_{LB}$ is required to rescue bacterial growth, we subsequently used the SMase-deficient $\Delta sph$ mutant for spike-in experiments in $MOD_{Val*}$ (Fig. 6f, g). The $\Delta sph$ mutant was cultivated in the presence of SMase, $EVs_{LB}$, or both in combination. As expected, the $\Delta sph$ mutant was unable to utilise $EVs_{LB}$ to rescue the growth impairments observed in $MOD_{Val*}$. It is only through the introduction of external SMase that the valine deficiency in $MOD_{Val*}$ can be compensated for and growth impairments overcome (Fig. 6f, g).

In summary, these data demonstrate that SMase-mediated EV degradation is integral for bacterial nutrient acquisition and adaption.

## Discussion

Conventionally, bacterial EVs have been considered stable vehicles for delivering molecular cargo, including virulence factors and signalling molecules to cells[12,14,37–39]. Our study calls into question this prevailing notion by demonstrating that EVs can undergo active degradation under particular environmental conditions. This process is orchestrated by SMase, which is co-secreted with the EVs. These findings expand the known repertoire of bacterial strategies for resource management and adaption, thereby adding a dynamic layer to the regulation of EV populations in fluctuating environments.

The inherent value of EVs as a nutrient source is attributable to their biomolecular composition, which encompasses lipids, proteins

and nucleic acids. It is plausible that bacteria have evolved mechanisms to utilise these resources in nutrient-deprived environments. As revealed by our work, targeting the lipid bilayer of EVs by a lipid-cleaving enzyme, such as SMase, emerges as a particularly efficacious strategy to increase the availability of EV nutrients for bacterial growth.

Utilising *B. cereus* and *S. aureus* as model organisms, we demonstrated that EV degradation is an enzyme-mediated process that stresses the importance of the interplay between EVs and their environmental milieu. As SMase production is a common feature among pathogenic bacteria, including *Clostridium perfringens*, *Listeria monocytogenes*, and *Staphylococcus epidermidis*[40], we hypothesise that EV-based SMase-mediated nutrient acquisition is not restricted to *B. cereus* and *S. aureus*, but can be applied to SMase-producing bacterial species in general. However, an additional prerequisite for EV-based SMase-mediated nutrient acquisition is linked to the EV architecture. As evidenced by the findings of our work, only EVs enriched in sphingomyelins can be degraded by SMase. SMase is a hydrolytic lipid-cleaving enzyme that cleaves sphingomyelin to phosphocholine and ceramide[41]. Sphingomyelins, alongside glycosphingolipids and ceramides, belong to the lipid class of sphingolipids[42] and are major constituents of biological membranes[43].

As demonstrated in this work, the lipid architecture of $EVs_{LB}$ dictates their stability and interaction with SMase. Based on our data, we propose the following model for SMase-mediated EV nutrient acquisition, depicted in Fig. 7: SMase-producing bacteria utilise sphingomyelin-rich EVs as recyclable nutrient reservoirs to boost growth when nutrients become exhausted. In contrast, EVs lacking sphingomyelins cannot undergo degradation, leaving their nutrients inaccessible to bacterial cells.

Our proposed model follows socioeconomic principles: rapidly growing bacteria in particular nutrient-rich environments can store nutrients in EVs for later use. When nutrients become exhausted, EVs are scavenged to boost growth and may also support bacterial adaption to changing environments. As such, EVs may serve as recyclable nutrient reservoirs. This hypothesis is supported by our finding that the depletion of EVs from conditioned LB significantly reduces growth, as well as by the results of the differential proteome analysis, which revealed the transient presence of the putative carbon starvation protein A in $EVs_{LB}$ at 9 h but not in $EVs_{LB}$ at 15 h. On the contrary, in MOD and RPMI, generally fewer nutrients are available, which is reflected in slower bacterial growth. Thus, it is tempting to speculate that $EVs_{MOD}$ have different functions, such as mediating stress signals. The latter is supported by $EVs_{MOD}$ proteomic data, which suggests that MOD-grown bacteria face translational and oxidative stress. The stress response reflected by the EVs is further supported by the increased abundance of ribosomal proteins and rRNA in $EVs_{MOD}$. Since bacteria lack strict compartmentalisation, they depend on a tightly regulated proteostasis to control protein production, folding, and transport. Exposure of bacteria to stressful and harsh environments results in an imbalance between transcription and translation[44–46]. Consequently, it is tempting to speculate that the loading of excess rRNA into $EVs_{MOD}$ enables the bacterial cell to fine-tune its translational landscape. However, further research is necessary to determine the potential role of EVs in translational regulation in bacteria.

As *B. cereus*, alongside the majority of other bacteria, is unable to synthesise sphingolipids de novo[47–49], it relies on active scavenging of these lipids from its environment, e.g. from eukaryotic host cells or complex laboratory media such as LB. While MOD medium and RPMI may therefore not be ideal for studying EV biology in a physiological context, they provide a controlled environment for harvesting EVs with consistent composition. Such a stable composition offers an advantage for the production of EVs used in clinical applications and the design of biomimetic vesicles[50] for industrial and environmental purposes[51]. Since lipids influence EV functionality in eukaryotic EVs[52–54], it is tempting to speculate that the bacterial EV lipidome also

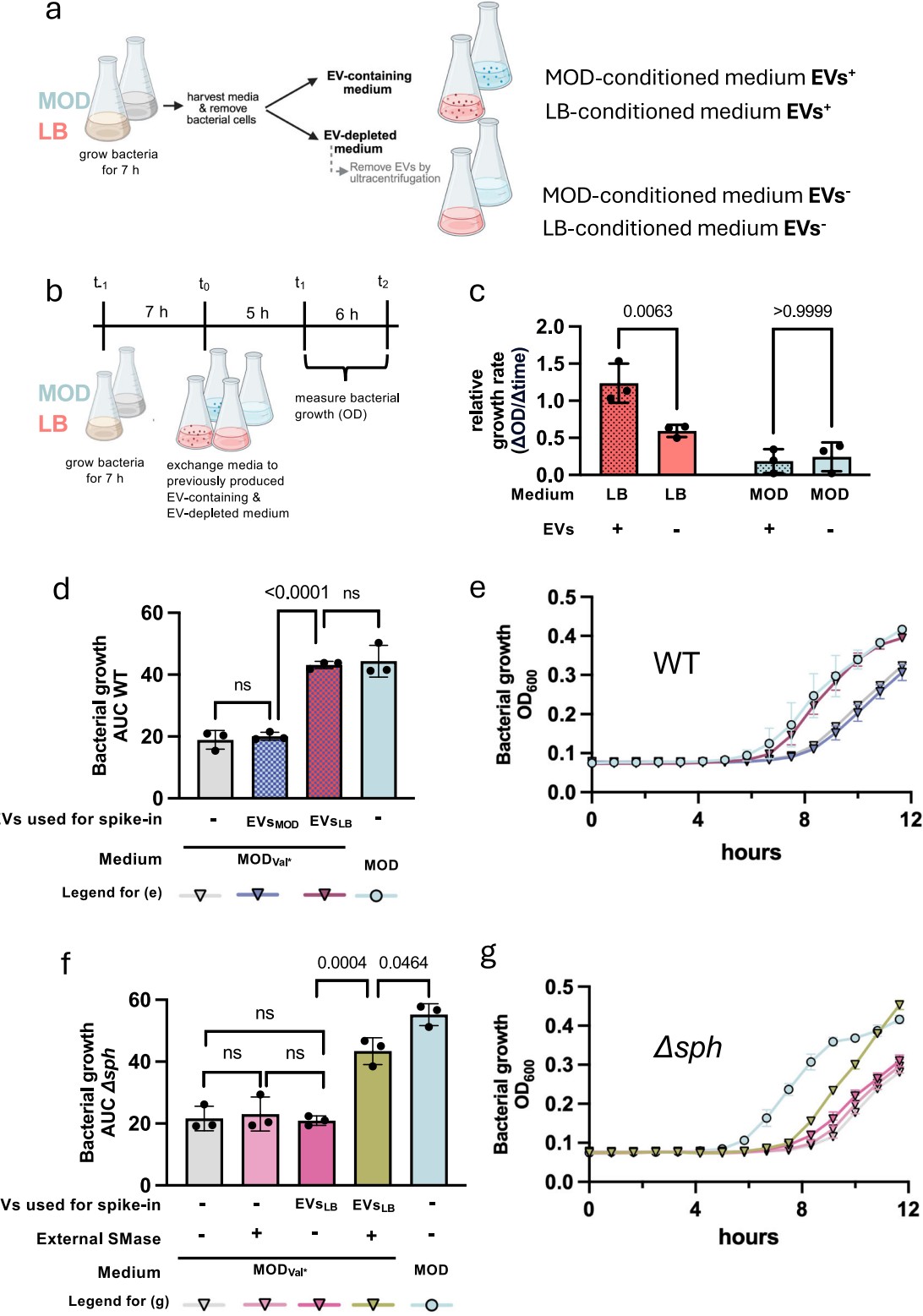

participates in different biological pathways. The specificity of SMase for sphingolipid-rich EVs highlights a previously underappreciated axis of environmental regulation, whereby bacteria tailor vesicle stability and fate through lipid remodelling.

The dynamic modulation of the EV lipidome in response to environmental cues, coupled with SMase secretion, enables bacteria to fine-tune vesicle stability and resource availability. This mechanism

likely confers a selective advantage in competitive or fluctuating environments, contributing to bacterial fitness and survival.

From an evolutionary perspective, extracellular nutrient storage in EVs provides a mechanism for bacterial populations to buffer transient nutrient limitation, extending beyond the confines of the individual cell. By this mechanism, bacterial EVs can support not only the producing cell but also neighbouring bacteria sharing the same

**Fig. 6 | EV-mediated nutrient release requires SMase-mediated EV degradation.** The impact of EVs on bacterial growth and the potential recycling of EV components as nutrients was assessed. **a** First, cell-free conditioned media were produced by growing bacteria for 7 h in LB and MOD. The media were harvested, sterile-filtered and split into two parts. One part was ultracentrifuged to remove EVs (EVs⁻) while the other part was not manipulated and thus still contained endogenous EVs (EVs⁺). **b** Next, bacterial cultures were grown for 7 h in LB and MOD, and media were exchanged to the previously produced culture media (**a**). **c** Then, bacterial growth was monitored by measuring the $OD_{600}$ 5 h ($t_1$) and 11 h ($t_2$) after the medium exchange. The relative growth rate ($\Delta OD_{600}/\Delta time$) was compared between media containing EVs (EVs⁺) and media without EVs (EVs⁻). Data presented stem from three independent experiments. **d–g** The valine auxotrophy of *B. cereus* F4810/72 was leveraged to investigate the recycling of EV components by bacteria using an automated growth recorder. All experiments were conducted three times independently, with four technical replicates each. MOD containing minimal amounts

of valine ($MOD_{Val*}$), allowing bacteria to grow after a prolonged lag phase (**e, g**), was prepared and used to investigate the growth of bacteria in $MOD_{Val*}$ with and without spike-in of EVs compared to MOD. **d, e** WT *B. cereus* was grown in $MOD_{Val*}$, in $MOD_{Val*}$ spiked-in with equal amounts of EVs$_{LB}$ or EVs$_{MOD}$, and MOD without spike-in of any EVs. Growth was monitored over 12 h. Areas under the curve (AUCs) were calculated (**d**) and the respective growth curves are presented (**e**). **f, g** The isogenic SMase-deficient mutant Δ*sph* was grown in $MOD_{Val*}$, in $MOD_{Val*}$ supplemented with commercial *B. cereus* SMase, in $MOD_{Val*}$ spiked-in with EVs$_{LB}$ or in $MOD_{Val*}$ spiked-in with EVs$_{LB}$ and commercial *B. cereus* SMase, and in MOD without additional spike-in of EVs. Experimental settings and growth analyses are the same as in (**d, e**). Statistical significance was calculated by one-way ANOVA with Bonferroni's multiple comparison test (**c, d, f**). *$p < 0.05$, **$p < 0.01$, ***$p < 0.001$, ****$p < 0.0001$. **c–g** Data presented reflect mean ± SD. **a, b** Created in BioRender. Ehling-Schulz, M. (2026) https://BioRender.com/jiyj53t. Source data are provided as Source Data file.

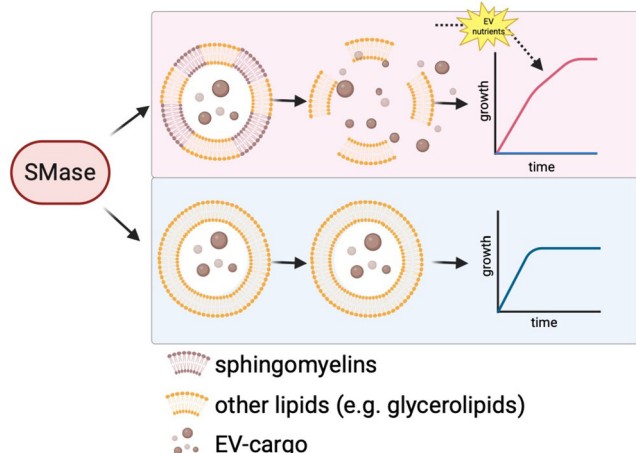

**Fig. 7 | EVs as recyclable nutrient reservoirs: proposed model of SMase-mediated EV nutrient acquisition.** In nutrient-complex environments, such as LB and hfRPMI, SMase actively degrades bacterial EVs due to their specific lipid architecture, characterised by the presence of sphingomyelins (^SM+EVs). As SMase can target ^SM+EVs, the released EV cargo serves as a nutrient boost, promoting bacterial proliferation when nutrient resources are exhausted. On the contrary, in defined nutrient-scarce environments, such as MOD and RPMI, bacterial EVs exhibit resistance to SMase-mediated degradation due to the lack of sphingomyelin in their membranes (^SM-EVs). Without its target, SMase cannot degrade such EVs. Consequently, ^SM-EVs remain intact, thus preventing their utilisation as recyclable nutrient reservoirs for bacterial cells when nutrients become limited. Created in BioRender. Ehling-Schulz, M. (2026) https://BioRender.com/938adhd.

environment, thereby promoting collective survival under nutrient-limited conditions. Such EVs may act as 'selectively accessible packages', where only SMase-secreting bacteria, such as *Clostridium perfringens*, *Listeria monocytogenes*, and *Staphylococcus epidermidis*[40], can access the stored nutrients, adding a layer of specificity within microbial communities. However, further research will be required to explore the extent to which this extracellular nutrient storage can be used as a public good by the entire bacterial community or whether it confers a selective advantage for the producing bacteria or particular members of the bacterial community. Extracellular storage inevitably carries the disadvantage of potential loss through diffusion or environmental clearance. Yet this diffusion can also be advantageous, e.g. by facilitating host-pathogen interactions by systemic dissemination of EVs[11,55] or priming distant niches, similar to mechanisms described for EVs in cancer metastasis[56]. By contrast, intracellular storage allows rapid access to nutrients but restricts benefits to single cells,

highlighting a trade-off between single-cell and community-level resource management strategies.

SMase-mediated EV degradation also reframes current concepts of EV functionalities. EVs protect, package, and deliver a wide range of bioactive molecules[14,37,57,58]. By actively modulating EV turnover, SMase impacts EV stability and specificity, and may also affect quorum sensing[37], toxin delivery[14], nutrient acquisition[58] and DNA transfer[59]. Moreover, SMase-mediated EV degradation likely impairs EVs' ability to act as decoys against phages and antibiotics[60,61]. Deciphering the precise contribution of SMase-mediated EV degradation to the functional repertoire of EVs will be crucial to understanding how EV functionality is integrated into the regulatory networks that shape bacterial communities.

The ability to modulate EV turnover in response to nutrient availability may influence inter-bacterial competition, community structure, and even host-pathogen dynamics. From a translational perspective, targeting SMase activity or manipulating EV lipid composition could offer novel approaches for antimicrobial intervention or microbiome engineering. Therefore, further insight into bacterial EV heterogeneity and the distribution of lipid species within EV populations is needed. While bacterial EV heterogeneity has so far been demonstrated mainly at the morphological level[62,63], it is plausible that different EV morphologies result from distinct EV lipid architectures, reflecting differences in membrane origin, curvature, or biogenesis pathways[54,64,65]. Resolving EV lipid composition at the single-particle level will be crucial for determining whether all EV subtypes are equally susceptible to SMase-mediated degradation and for understanding how EV stability is regulated across heterogeneous populations. Emerging single-vesicle analytical approaches, such as direct stochastic optical reconstruction microscopy (dSTORM), Raman trapping analysis and single EV flow cytometry, allow analysing single EVs in eukaryotic systems[66–68], but their applicability remains very limited for bacterial EVs due to the absence of specific marker proteins and the high diversity of bacterial species[69]. Adapting and extending these technologies to bacteria will be essential to determine how bacterial communities utilise EV heterogeneity to fine-tune EV stability, cargo delivery and resource allocation, thereby offering further mechanistic insights into SMase-mediated EV degradation.

In summary, our study reveals a previously unrecognised, environmentally responsive mechanism of bacterial EV turnover, governed by the interplay between vesicle lipid composition and secreted SMase activity. This process enables bacteria to dynamically regulate EV populations and recycle nutrients, linking EV biology with broader metabolic and survival strategies. These insights advance our conceptual framework for bacterial EV function and pinpoint new directions for research into microbial adaption and intervention.

## Method

### Bacterial strains

Experiments in this study were conducted using the emetic B. cereus reference strain F4810/72[70] (aka AH187) and an isogenic SMase knockout mutant ($\Delta sph$) of *B. cereus* F4810/72[30]. For complementary experiments, the *B. cereus* strains BC435, F588/94, and WBSC10925, as well as the *Staphylococcus aureus* strain Newman, were used.

### Growth media, production of conditioned media and EV-depleted conditioned media

The composition of the growth media LB[71], MOD[72]. and MOD$_{VAL*}$ is detailed in Supplementary Data 6. The host-factor-enriched RPMI (hfRPMI) was produced by growing Caco-2 cells in RPMI (Thermo Scientific, Waltham, MA, USA), supplemented with 10% FBS (Thermo Scientific, Waltham, MA, USA), for 21 days as described earlier[73]. Subsequently, this hfRPMI was subjected to ultracentrifugation overnight at $180,000 \times g$ to deplete the medium from Caco-2 EVs and then frozen until further use.

Ultracentrifuged sterile LB (LB*) was prepared by sterile-filtering LB (0.45 μm) and ultracentrifugation for 16 h at $180,000 \times g$. To obtain 'cell-free conditioned media', bacteria were grown for 7 h at 30 °C while shaking (120 rpm) in the respective media and the supernatants were cleared of bacterial cells by a series of centrifugation steps: $3000 \times g$, $4000 \times g$, 4 °C, 15 min each, followed by filtration (pore size 0.45 μm), and centrifugation at $10,000 \times g$, (4 °C, 15 min). To further produce 'EV-depleted cell-free conditioned media', cell-free conditioned media were ultracentrifuged overnight (16 h, $180,000 \times g$).

### Production and isolation of EVs

For the production of EVs, bacteria were grown in the respective media at 30 °C, the commonly used *B. cereus* growth temperature, and 120 rpm shaking as described previously[74]. The cultures were then harvested at the indicated time points. To isolate EVs, the 100 ml culture was centrifuged ($3000 \times g$ for 15 min; at 4 °C; $4000 \times g$ for 15 min; at 4 °C). Supernatants were filtered through a 0.45 μm filter (Merck Millipore, Darmstadt, Germany), and the filtrate was centrifuged at $10,000 \times g$ for 15 min at 4 °C. Subsequently, EVs were pelleted from this supernatant via ultracentrifugation at $180,000 \times g$ for 1 h using Opti-Seal Polypropylene tubes (MLA-50 rotor, Optima MAX-XP, Beckman-Coulter, Brea, CA, USA). The supernatant was discarded, and the pellet was resuspended in 30 μl PBS (pH 7.4) for every 100 ml of bacterial liquid culture.

### Density gradient ultracentrifugation of EVs

EVs were further purified using a density gradient (DG) of 12% to 60% OptiPrep (Iodixanol; Sigma-Aldrich, St. Louis, MO, USA), as described previously[19]. OptiPrep was diluted in PBS (2 ml/fraction), $5.6 \times 10^9$ EVs/ml were loaded on top and subsequently subjected to ultracentrifugation at $180,000 \times g$ for 16 h at 4 °C (MLA-50 rotor, Optima MAX-XP, Beckman-Coulter, Brea, CA, USA). The single fractions were collected, diluted with PBS to 15 ml and the OptiPrep reagent was removed by Amicon filter columns (100 kDa cut-off, Sigma-Aldrich, St. Louis, MO, USA). Each fraction was centrifuged twice for 30 min at $10,000 \times g$, at 4 °C. The resulting concentrated sample was diluted to a final volume of 300 μl PBS and analysed for the particle number and size. Fractions containing EVs (F2–F4) were pooled and stored at −20 °C for further use.

### Characterisation of EVs

**Nanoparticle tracking analysis (NTA).** To quantify the amount and size of isolated particles, NTA was performed using a PMX-230 TWIN Laser ZetaView (Particle Metrix, Inning, Germany) with a temperature setting of 25.0 °C, with sensitivity and shutter set to 70. Recordings were analysed using ZetaView 8.04.02 software (Particle Metrix, Inning, Germany).

**Transmission electron microscopy (TEM) and atomic force microscopy (AFM).** Transmission electron microscopy was employed to visualise EVs, utilising either the drop-on-grid method or the preparation of ultra-thin sections of resin-embedded EVs. For the drop-on-grid method, EVs were resuspended to reach a final concentration of $2 \times 10^8/\mu l$, of which 10 μl were incubated for 60 min on formvar/carbon copper grids (300 mesh, FCF300H-CU-SB, Science Services, Munich, Germany). Samples were then fixed with 1% glutaraldehyde (5 min) in PBS, washed in ddH$_2$O, followed by negative staining with 2% aqueous uranyl acetate for 5 min. To prepare resin-embedded EV samples, EVs were pelleted and then overlaid with glutaraldehyde to preserve their morphology. Ultra-thin sections of EV samples were prepared as described previously[14]. Grids and resin-embedded ultra-thin *B. cereus* EV sections were examined using a transmission electron microscope (EM 900, Zeiss, Jena, Germany) equipped with a slow-scan CCD 2K-wide-angle dual-speed camera (TRS, Moorenweis, Germany).

To acquire AFM images, previously established protocols were followed[75,76]. In short, 5 μg of EVs were immobilised for 60 min in 100 μl PBS on 0.01% poly-L-lysine coated mica slides. The scanning was performed using a Dimension FastScan Bio (Bruker, Billerica, MA, USA) with a SCANASYST-FLUID+ tip (Bruker, Billerica, MA, USA) and a spring constant of 0.7 N/m. Before the start of the AFM measurement, the tip was calibrated using a mock-immobilised slide with PBS only. After scanning the sample, the images were processed using Gwyddion[77].

**Fourier-transform infrared spectroscopy (FTIR).** To assess compositional changes in EVs, FTIR spectroscopy was employed as described previously[14,78]. For the FTIR measurement, EVs sampled from three independent bacterial cultures were applied in technical quadruplicates to a 384-well silicon plate (Bruker Optics GmbH, Ettlingen, Germany). After drying the samples (40 °C, 20 min), FTIR spectra were recorded in transmission mode (spectral range: 4000–500 cm⁻¹; spectral resolution: 6 cm⁻¹)[79,80], using an HTS-XT microplate adaptor coupled with a Tensor 27 FTIR spectrometer (Bruker Optics GmbH, Ettlingen, Germany). For univariate analysis, technical replicates were averaged, subjected to vector normalisation and baseline correction. The subtraction spectrum of EVs$_{MOD}$-EVs$_{LB}$ was calculated based on 2nd order derivative spectra. In the obtained spectra data, regions corresponding to sphingolipids[81] (2800–3000 cm⁻¹, 1600–1680 cm⁻¹, 1260 cm⁻¹, 1455–1485 cm⁻¹, 1000–1110 cm⁻¹), glycerolipids[82] (1728–1742 cm⁻¹), and phospholipid vibrations[83] (1070–1260 cm⁻¹) were highlighted. For multivariate analysis, technical replicates were averaged, and then pre-processed by baseline correction, L2 normalisation, spline interpolation, smoothing with a Savitzky-Golay filter, and calculation of the second-order derivatives. For univariate processing of FTIR spectral data, OPUS software version 7.2 (Bruker Optics GmbH, Ettlingen, Germany) was used. Multivariate FTIR spectral processing, analysis and visualisation were performed by Python (version 3.11.0) using Pandas (version 2.2.3), Numpy (version 1.26.4), Matplotlib (version 3.8.4)[84], scipy (version 1.15.1)[85], and Scikit-learn (version 1.6.1)[86] libraries.

### Assessment of EV degradation

To assess EV degradation, different types of media were used: sterile media spiked with bacterial EVs, cell-free conditioned media containing endogenous bacterial EVs (produced as described above), and EV-depleted cell-free conditioned media spiked with bacterial EVs. To measure EV degradation, the media containing spiked-in EVs ($1.5 \times 10^9$ EVs/ml – $2.5 \times 10^{10}$ EVs/ml) or endogenous EVs were incubated at 30 °C and 120 rpm for 4.5 h, unless stated otherwise. NTA was used to determine the amount of EVs at the start of the experiment ($t_0$) and after 4.5 h or at the indicated time points.

To test the potential of exogenously added SMase to degrade EVs, *B. cereus* SMase (0.125 U/ml, Sigma-Aldrich, St. Louis, MO, USA) was added to sterile culture media spiked with *B. cereus* EVs. The following

substances were used to test whether EV degradation can be inhibited: Imipramine (700 µg/ml), Pepstatin (5 µg/ml), Aprotinin (14 µg/ml), PMSF (7 µg/ml), EDTA (150 mg/ml, all Sigma-Aldrich, St. Louis, MO, USA).

### SMase activity assay

SMase activity was measured in conditioned EV-depleted *B. cereus* media, for which bacteria were grown for 7 h at 30 °C. The Amplex® Red Sphingomyelinase Assay Kit (Waltham, MA, USA) was used to determine the production of sphingomyelinase, as described previously[30]. In brief, 10 µl of conditioned EV-depleted media was mixed with 1x reaction buffer, provided by the manufacturer, to reach a final volume of 100 µl. A working solution containing 2 U/ml horseradish peroxidase, 0.2/ml choline oxidase, 8 U/ml alkaline phosphatase and 0.5 mM sphingomyelin was prepared. Finally, 100 µl of the sample diluted in 1x reaction buffer was mixed with 100 µl of the previously prepared working solution in a black clear-bottom 96-well plate (Greiner, Kremsmünster, Austria). Fluorescence was measured after 30 min (excitation 530 nm; emission 590 nm).

### Sphingomyelin quantification assay

Sphingomyelin levels were quantified in EVs and in sterile bacterial culture media using a fluorometric sphingomyelin assay kit (Sigma-Aldrich, St. Louis, MO, USA), according to the manufacturer's instructions. In brief, EV protein content was determined, and samples were either adjusted to the indicated EV protein amount or to 15 µg total EV protein. EV samples were then diluted in the supplied assay buffer to a final volume of 50 µl per well. For analysis of bacterial culture media, 50 µl of media was used per well. A working reagent containing SMase reaction buffer and SMase working stock was prepared and mixed with the samples, following the manufacturer's instructions, and incubated for 2 h at 37 °C, protected from light. Subsequently, 50 µl of the sphingomyelin assay mixture (containing enzyme mix, red detection reagent, and assay buffer) was added to the samples and the plate was incubated for 1 h at room temperature. Subsequently, fluorescence was recorded (excitation 535 nm; emission 587 nm). A positive control supplied by the manufacturer (100 µM sphingomyelin) was included on each plate and assayed in parallel with a blank. Fluorescence values were corrected by subtracting the blank signal from all samples and controls.

### Quantitative proteomics using mass spectrometry

*B. cereus* EVs were analysed by quantitative proteomics using mass spectrometry at the Bavarian Centre for Biomolecular Mass Spectrometry (BayBioMS) at TU Munich. To obtain EVs, *B. cereus* cultures were grown in LB or MOD for 9 and 15 h, at 30 °C while shaking (120 rpm). EVs were collected by using ultracentrifugation, as described in the previous section. For sample preparation, the volume equivalent to 15 µg of EV protein was combined with 4 x Laemmli buffer (BioRad, Hercules, CA, USA), 2-mercaptoethanol (Sigma-Aldrich, St. Louis, MO, USA) and 10 x RIPA lysis buffer (Merck, Darmstadt, Germany). PBS was added to a final volume of 26 µl. To disrupt EVs, samples were frozen at −80 °C overnight. After thawing, samples were incubated at 70 °C for 5 min, followed by sonication for 5 min in an ice-cold water bath. For proteomic analysis, in-gel trypsin digestion was performed as described previously[87]. EV-proteins were separated on Nu-PAGE™ 4–12% Bis-Tris protein gels (Thermo Scientific, Waltham, MA, USA) for approximately 1 cm, after which the accumulated protein band was cut out, following reduction (50 mM dithiothreitol), alkylation (55 mM chloroacetamide), and digestion overnight with trypsin (Trypsin Gold, mass spectrometry grade, Promega, WI, USA). Samples were dried and subsequently resuspended in 25 µl of buffer A (2% acetonitrile, 0.1% formic acid in HPLC-grade water), of which 5 µl were injected for the mass spectrometry (MS) measurement. Liquid Chromatography (LC)−MS/MS measurements were carried out on a Dionex Ultimate 3000 RSLCnano system coupled to a Q-Exactive HF-X mass spectrometer (Thermo Fisher, Waltham, MA, USA). Injected peptides were delivered to a trap column (ReproSil-our C18-AQ, 5 µm, Dr. Maisch, 20 mm × 75 µm, self-packed) following transfer to an analytical column (ReproSil Gold C18-AQ, 3 µm, Dr. Maisch, 450 mm × 75 µm, self-packed) and separated using a 50 min gradient from 4 to 32% of solvent B (0.1% FA, 5% DMSO in acetonitrile) in solvent A (0.1% FA, 5% DMSO in HPLC grade water) at 300 nl/min flow rate. The Q-Exactive HF-X mass spectrometer was operated in data-dependent acquisition and positive ionisation mode, as described previously[87]. Using Max-Quant software 1.6.3.4[88] with its integrated Andromeda search engine[89], peptides were identified, quantified and matched by the MS2 spectra matched against the *B. cereus* AH187 proteome from Uniprot (UP000002214), along with common contaminants.

Trypsin/P was selected as the proteolytic enzyme, and carbamidomethylated cysteine was set as a fixed modification. Variable modifications included methionine oxidation and protein N-terminal acetylation. The FDR—set to 0.01 for peptide spectrum matches and proteins—was obtained using a target-decoy method with reversed protein sequences. LFQ intensities were subsequently analysed using Perseus version v2.0.10.0[88]. To identify proteins that were consistently detectable across different media and time points, we filtered the data using the function 'filter rows based on valid values' set to '50% in at least one group'. Missing values were handled through imputation based on a normal distribution, using Perseus's default parameters (width = 0.3, down shift = 1.8). The imputed dataset was normalised by quantile normalisation before statistical testing. Volcano plots were created by plotting the $\log_2$FC against the $-\log_{10}$ adjusted p-values for each pairwise comparison. PCA was obtained using LFQ intensities and the AMICA platform[90] (https://bioapps.maxperutzlabs.ac.at/app/amica).

The proteomics data have been deposited in the ProteomeXchange Consortium with the following identifier PXD065751.

**Enrichment analysis**. Gene ontology and enrichment analyses were performed using ShinyGO v0.80 (http://ge-lab.org/go/), with the reference organism *B. cereus*[91]. Significantly enriched GO terms with adjusted p-value with false discovery rate (FDR) < 0.005 were included for further analysis. Python plotly and goatools libraries were used for GO term redundancy removal and visualisation using a Sankey diagram.

To reduce redundancy, redundant GO terms were removed using hierarchical filtering via the goatools library, which leverages the GO-directed acyclic graph (DAG) structure from the go-basic.obo ontology to retain only the most specific (child) terms.

To refine the GO terms, a semantic similarity-based filtering step using a weighted Jaccard index and gene annotation-derived information content (IC) was applied for each condition[92]. Pairs of terms exceeding the similarity threshold (threshold = 0.7) were filtered, retaining only the term with the higher information content (IC) from each pair. The filtering was performed independently for each condition. To validate the robustness of the weighted Jaccard approach, the results were compared with those produced using Resnik and Lin similarity metrics, yielding comparable reductions. For visualisation of condition-specific enrichment and flow patterns, Sankey diagrams were implemented to map the relationships. This layered design facilitates the inspection of both shared and condition-specific pathway–gene relationships. To perform HCA analysis and to illustrate the heat map, the following tool was used[93]: http://www.heatmapper.ca. To obtain hierarchical clusters, the following settings were used: Clustering Method: Average Linkage and Distance Measurement: Kendall's Tau. The proteins depicted in the heat map were chosen based on GO terms. Stress response proteins were identified by filtering for proteins with child-terms of the 'response to' GO-term.

## Transcriptomics

To isolate RNA from EVs, for each sample 100 ml of bacterial culture was prepared by growing bacteria for 9 h at 30 °C while shaking (120 rpm). EVs that were collected by ultracentrifugation from the entire culture volume (100 ml) were used, and RNA was isolated using the Monarch® Total RNA Miniprep Kit (New England BioLabs, Ipswich, MA, USA). Instructions for tough-to-lyse samples, such as bacteria, yeast, and plants, were followed. For the library preparation, the CORALL RNA-Seq V2 kit (Lexogen, Vienna, Austria) was used, following the protocol for short insert sizes. No rRNA depletion or mRNA enrichment was performed prior to preparing this library. The size distribution of the final libraries was evaluated via capillary electrophoresis using the DNA 1000 Assay on the Bioanalyzer 2100 (Agilent Technologies, Santa Clara, CA, USA). Sequencing was performed using 100 sequencing cycles on a NovaSeq 6000 platform (Illumina, San Diego, CA, USA) with a single-end 150 bp (SE150) format. The genomes of AH187 and NVH0075-95 served as reference genomes. Prior to alignment, all reads were trimmed to remove adaptor sequences. Both before and after trimming, sequencing quality was assessed using FastQC[94]. The trimmed reads were then aligned to the reference genome using Bowtie's "best" algorithm[95]. The resulting Bowtie output was processed to evaluate read counts using HTSeq with the intersection_nonempty mode[96].

The transcriptomics data have been deposited to European Nucleotide Archive with the following accession number PRJEB94405.

## Quantitative lipidomics using mass spectrometry

The lipidome of *B. cereus* EVs was analysed by quantitative lipidomics via mass spectrometry at the Metabolomics Core Facility at EMBL Heidelberg. For the EV lipid extraction, *B. cereus* was grown in LB or MOD for 9, at 30 °C while shaking (120 rpm), and EVs were collected via ultracentrifugation, as described in the previous section. To extract EV-lipids, equal amounts of EVs ($1.2 \times 10^9$) were pelleted and resuspended in 150 μl of ice-cold isopropanol (Carl Roth, Karlsruhe, Germany) containing internal standards (provided by the EMBL Metabolomics Core Facility). The samples were incubated on ice for 20 min, followed by 5 min of vortexing and centrifugation at $14,000 \times g$ for 15 min at 4 °C. The supernatant containing the extracted lipids was subsequently frozen until further analysis. A quality control (QC) sample was prepared by pooling 20 μl from each sample together. LC-MS/MS analysis of the extracted lipids was performed on a Vanquish UHPLC system coupled to a Q-Exactive plus HRMS (Thermo Scientific, Waltham, MA, USA) in both ESI positive and negative mode[78]. In brief, lipid and fatty acid separation was carried out on a Kinetec C18 column ($1 \times 100$; 2.6 μM; flow rate 0.26 ml/min; 30 °C). Solvents were buffered either with 10 mM ammonium formate (positive mode) or 10 mM ammonium acetate (negative mode). Solvents were further supplemented with 0.1% formic acid in isopropanol:acetonitrile (9:1) for mobile phase A, and 0.1% formic acid in water:acetonitrile (6:4) for solvent B. Lipid detection was carried out with an HRMS full scan at a mass resolving power of $R = 70,000$ and a range of 200–1500 m/z. Data-dependent MS/MS scans were collected using full scans with higher energy collisional dissociation (HCD) at a mass resolving power of $R = 17,500$. MS parameters in the Tune software were set as follows: spray voltage, 3200 V; capillary temperature, 280 °C; probe heater temperature, 300 °C; sheath gas, 30 units; auxiliary gas, 5; S-Lens RF level, 60 units. During data acquisition, data-dependent tandem mass spectra were obtained for the top 10 most intense precursors. Samples were randomised for LC-MS/MS analysis sequence, and the pooled quality control samples, along with the blank samples, were injected at the beginning of the sample analysis sequence to stabilise the LC-MS/MS system. A quality control sample was injected after every six samples to monitor the instrument's stability (%CV < 20). The chromatograms were evaluated, and the stability of quality control samples was checked with PCA plots before proceeding with further data

analysis. Thermo Xcalibur software (Thermo Scientific, Waltham, MA, USA) was employed to acquire data and perform preliminary data evaluation, including assessment of chromatogram quality and obtaining extracted ion chromatograms, peak integration and raw data visualisation. Alignment of LC-MS/MS data, peak picking, adduct deconvolution, and normalisation were done using Progenesis QI (Waters, Nonlinear Dynamics, Newcastle upon Tyne, UK) software. Lipid annotations were performed using LipidBlast database and EMBL Metabolomics Core Facility spectral library (http://curatr.mcf.embl.de) for MS and MS/MS-based identification. The mass tolerance in MS1 and MS2 was 5 ppm and 10 ppm, respectively.

To analyse differentially expressed lipids and lipid abundance, files from both negative and positive measurement modes were combined to avoid counting duplicates or lipids that are measured in both modes. Next, the average of all quality control (QC) samples was calculated for each lipid. Then, we compared QC lipid values between positive and negative modes, selecting only the mode with the higher value for further analysis. Duplicates were removed by keeping the replicate with the higher value. Finally, both modes were combined into a single list, ensuring each lipid had only one respective measurement. To annotate the lipid class and subclass, the metabolomics workbench, available at www.metabolomicsworkbench.org[97] was used. To determine the number of abundant lipid species, we considered lipids to be abundant in a replicate only if their detected value exceeded the blank. With three biological replicates analysed per condition, lipids were only counted as abundant if present in at least two out of three biological replicates. Subsequently, we quantified the relative proportions of each lipid class and subclass.

The lipidomics data have been deposited to MetaboLights repository with the study identifier MTBLS13260[98].

## Reactive oxygen species assay

To assess the accumulation of ROS in bacterial cells, the CellROX™ Orange Reagent (Thermo Scientific, Waltham, MA, USA) was utilised following the manufacturer's protocol with minor modifications. Briefly, bacterial cells were cultivated for 15 h in either 100 ml LB or MOD at 30 °C. From this culture, 1 ml was used to assess ROS accumulation. After pelleting ($3000 \times g$, 15 min, room temperature (RT)), the bacterial pellet was washed once with 0.9% NaCl, following incubation with CellROX™ Orange Reagent (10 μM) at 37 °C for 60 min. An unstained control sample was prepared to account for autofluorescence. After staining, the cells were washed three times with 0.9% NaCl, spotted on a slide and dried overnight. Imaging was performed using a bright-field and wide-field fluorescence microscope (Carl Zeiss, Oberkochen, Germany) with a deep orange filter, and Zeiss ZEN Blue software was applied for image acquisition. The mean fluorescence intensity of the acquired images was assessed using ImageJ[99]. Images were converted to 8-bit formats to enable thresholding of signal intensities in regions of interest (ROI).

## Zymography

EV-depleted cell-free conditioned LB derived from WT and Δ*sph* bacteria was tenfold concentrated using Amicon spin-filters (10 kDa, Sigma-Aldrich, St. Louis, MO, USA). Of this concentrate, 1 μl was loaded onto a 12.5% SDS-PAGE gel supplemented with 1% casein or 1% gelatine (both Sigma-Aldrich, St. Louis, MO, USA). Proteins were separated for 2 h at 120 V, with a maximum of 20 mA at 4 °C. Thereafter, gels were regenerated using 2.5% Triton X-100 and subsequently equilibrated with the first 5 mM CaCl$_2$, 50 mM Tris (pH 7.4; 20 min; Carl Roth, Karlsruhe, Germany) and the second 0.2 M NaCl, 10 mM CaCl$_2$, 50 mM Tris, 0.02% Brij35 (pH 7.4; 20–30 h; Carl Roth, Karlsruhe, Germany). The gel was stained overnight at RT using PageBlue Protein staining (Thermo Scientific, Waltham, MA, USA) and destained with ultrapure water[100].

## Assessing the effect of EVs on bacterial growth dynamics

Bacterial growth in LB and MOD media was analysed by modelling growth dynamics using a modified Gompertz[101] function:

$$OD_{600}(t) = A^* \exp\left\{-\exp\left[\frac{\mu_m^* e}{A}(\lambda - t) + 1\right]\right\} \quad (1)$$

$A$ is the maximum $OD_{600}$, $\mu_{max}$ is the maximum specific growth rate, $\lambda$ is the lag time, and $e$ is Euler's number. Initial parameters were derived from raw data and refined using the Solver tool in Microsoft Excel, which minimised the sum of squared differences between the observed OD values and the predictions of the Gompertz model.

To assess the impact of EVs on bacterial growth, cell-free conditioned LB and cell-free conditioned MOD media were prepared as described above. While one part of each media was left unmanipulated (EVs+), the other part was subjected to EV-depletion (EVs−) in order to remove endogenously produced EVs. The following day, bacteria were grown for 7 h at 30 °C, 120 rpm (100 ml, Erlenmeyer flasks) and subsequently subjected to media exchange ($t_0$). Therefore, bacteria were harvested by centrifugation (5000 × $g$, 3 min, RT), resuspended in previously prepared pre-warmed EV-containing conditioned medium (EVs+) or EV-depleted conditioned medium (EVs−) and further incubated at 30 °C, 120 rpm. Growth of the bacteria was measured as $OD_{600}$ and the relative growth rate (average change in $OD_{600}$/hour) was calculated between $t_1$ (5 h) and $t_2$ (11 h) after medium exchange at $t_0$.

An automated bacterial growth curve recorder (Bioscreen G Pro, Bio Laboratories, Singapore) was used to investigate the recycling of EV components by bacteria. Bacteria (either *B. cereus* F4810/72 WT or its isogenic Δ*sph* mutant) were inoculated in technical quadruplicates in the respective culture media in 100-well honeycomb plates (Thermo Scientific, Waltham, MA, USA) and grown for 12 h at 30 °C with orbital shaking. For spike-in experiments, equal amounts of EVs_LB or EVs_MOD (1.7 ×10⁹ EVs/ml) were used and 0.125 U/ml of *B. cereus* SMase (Sigma-Aldrich, St. Louis, MO, USA) were added. Bacterial growth was measured as $OD_{600}$ every 5 min and every 10th data point is shown for visualisation. The area under the curve (AUC) was calculated using the GraphPad Prism (Prism 10 for macOS, Version 10.4.2) plug-in.

## Statistical analysis

Statistical analysis was performed with Prism 10 for macOS, Version 10.4.2. One-way or two-way ANOVA followed by Bonferroni's multiple comparisons test was used when more than two groups were compared. An unpaired two-tailed *t*-test was used when comparing two groups. All graphs display individual data points and the mean value with standard deviation (SD). Details of the statistical tests used are shown in the respective figure legends. *p* values smaller than or equal to 0.05 were considered significant.

## Reporting summary

Further information on research design is available in the Nature Portfolio Reporting Summary linked to this article.

## Data availability

The proteomic data generated in this study have been deposited in the ProteomeXchange Consortium via the PRIDE database under accession code PXD065751. The lipidomic data generated in this study have been deposited in the MetaboLights database under accession code MTBLS13260. The transcriptomic data generated in this study have been deposited in the European Nucleotide Archive database under accession code PRJEB94405. Source data are provided with this paper.

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

## Acknowledgements

We thank all laboratory members for their helpful discussions. In addition, we are grateful to Raphaela Wahrmann for her technical assistance and to Georg Csukovich. Additionally, we want to acknowledge Viktoria Krey for providing the *B. cereus* SMase (*sph*) knock-out mutant strain and Tom Grunert for providing the *S. aureus* Newman strain. This research was supported using resources of the VetCore Facility (VetImaging) of the University of Veterinary Medicine Vienna. We acknowledge the support of the EMBL Metabolomics Core Facility (MCF) in the acquisition and analysis of liquid chromatography-mass spectrometry data. This work was supported by the Vienna Anniversary Foundation for Higher Education (H-409332/2021) to A.L.D., the OEAD fellowship "Aktion Österreich-Tschechien" (DZS/2024/00070) to A.L.D., as well as the CIISB infrastructure grant #240075 to A.L.D. The authors acknowledge

funding from CIISB, Instruct-CZ Centre, supported by MEYS CR grant numbers LM2023042 to J.P. and LUC24105 to J.P. and European Regional Development Fund-Project No. CZ.02.01.01/00/23_015/0008175 to J.P. The work was supported by ERASMUS+ Student Mobility for Traineeships (SMT) to T.V.E. This research was funded in part by the Swiss National Science Foundation (SNSF) Sinergia grant CRSII5_209253 to M.E.S. The Q Exactive HFX mass spectrometer of the BayBioMS was funded in part by the German Research Foundation (INST 95/1435-1 FUGG) to C.L.

## Author contributions

A.L.D.: writing—original draft, visualisation, validation, methodology, investigation, formal analysis, funding acquisition, conceptualisation; T.E.: writing—original draft, visualisation, validation, methodology, investigation, formal analysis, conceptualisation; M.A.: methodology; M.Y.: methodology, D.C.: methodology; B.K.: methodology; S.W.: methodology; W.T.: methodology; I.W.: methodology; S.K.: methodology; C.L.: methodology; J.P.: methodology, resources; M.P.: resources, supervision, validation, conceptualisation; M.E.S.: writing—review & editing, validation, supervision, resources, project administration, investigation, funding acquisition, conceptualisation.

## Competing interests

The authors declare no competing interests.
