## [Transparent Peer Review file · Nature Communications]

Bacterial extracellular vesicles as recyclable nutrient reservoirs

Corresponding Author: Professor Monika Ehling-Schulz

Version 0:

Reviewer comments:

Reviewer #1

(Remarks to the Author)

In this study the authors present novel data demonstrating that bacteria can utilize extracellular vesicles as a nutrition source under certain conditions. The ability to utilize EVs as a nutrition source is dependent on two factors (i) the lipid composition of the EV membrane and (ii) the ability of the bacteria to secrete an enzyme that degrades the EV, thus releasing its contents. Overall the experiments are well performed and appropriate controls are included. There are a number of places where additional controls would have increased rigor of the study and I have outlined those below. The idea that EVs may function as a nutritional storage system is novel and the data presented support the hypothesis that this may be possible under certain circumstances. Since this is a novel hypothesis I feel it's important the authors provide a more detailed discussion about how this impacts our knowledge of bacterial communities and the role of EVs (see comment below).

The title of the manuscript is somewhat vague and cryptic. Perhaps consider rephrasing "provides new clues on bacterial vesicle function" with something like "suggests a role for EVs in nutrient storage/recycling"?

Line 107: "...were found to be larger than EVsMOD". Please be specific. How much larger were they?

The results shown in supplemental fig 2d and e are important to the study and should be moved to the main manuscript.

Line 144: "The dynamics of bacterial EV production were monitored...". Can the authors clarify if EV production, EV degradation, or both was measured in this assay?

Line 178: "To assess whether EV degradation manifests in the particular proteomes..." Please rephrase/clarify the meaning here, it is unclear.

Line 165: When FTIR spectroscopy is first introduced in the manuscript it would be useful to include a sentence or two outlining what this technique is used for to assist readers who are not familiar with the approach.

Line 173: Fig 1g should be Fig 1i

Line 178: "To assess whether EV degradation manifests in the particular proteomes..." Please rephrase this sentence as the meaning is unclear.

Line 188: "The volcano plot visualises exclusive enrichment of proteins at 9 h in EVSLB (Supplementary Fig. 4a-d), further supporting the hypothesis that EVsLB degrade over time (Supplementary Fig. 4a)". I don't see how the data presented in volcano plots supports these statements. The data in Supplementary Fig. 4a seems to indicate a roughly equivalent number of proteins are more abundant at 15 hours and 9 hours. Can the authors clarify?

Please explain the origin of the data in figure 2D in more detail. The figure legend says it was constructed using differentially regulated proteins from the comparison presented in supplementary Figure 4 which helps, however only minimal details are given in the manuscript text. Please also explain what is meant by "The enrichment of non-redundant pathways" on line 192

In several places throughout the manuscript the term “plain LB” is used. Does this refer to sterile unused LB? If so please clarify and use the term “sterile LB” instead.

An important control is missing from Fig 4B. The SMase inhibitor (imipramime) should have been added along with exogenous SMase to show specific inhibition of SMase activity restores EV degradation.

Line 257: “Various protease inhibitors tested in parallel did not inhibit this temperature-dependent, enzyme-driven process as effectively as the SMase inhibitor Imipramine.” This statement is misleading and is not supported by the results in supplemental figure 6b. Results in supplemental fig 6b show that addition of proteases did significantly inhibit EV degradation and in some cases (aprotinin) this inhibition appears to be comparable to or more than that observed with SMase inhibitor (imipramime). This needs to be explained/discussed in more detail. The iss

Fig 4 a, c, and e. The schematics for each experiment are actually somewhat confusing and don't really add to the figure. At the bottom of each diagram the symbols used are depicted presumably as a “key” for the figure. However, at first it appears that they are part of the schematic figure itself. Perhaps enclosing them in a box would make this more obvious and help with clarity of the figure. Fig 4e in particular is not helpful as It doesn't outline the experiment well (should indicate which is the deltasph mutant).

The result in supplemental fig 6f is quite important to the overall study and should perhaps be moved to the main manuscript.

Fig 5a and b. Could the authors provide this data as a plot of growth (OD600 Vs time) as supplemental data as this is the format most researchers are familiar with for bacterial growth.

Line 297: “we screened the proteome data...” What proteome data are they referring to? Is it the data generated in this study (depicted in Fig 2?). If so this needs to be clearly stated. Also, how was the data screened? More details need to be provided.

Line 299: “(child-terms of the ‘response to’ GO-term Supplementary Table 4)”. What is meant by this statement? The meaning is unclear.

Line 309: “the exclusive enrichment of Dps1, Dps2 and the putative carbon starvation protein A in EVsLB indicates that bacteria cultivated in LB experience nutritional stress.” Can the authors clarify whether they believe EV cargo proteins are present simply because that are simultaneously abundant in the cytoplasm of the originating cell, or if they believe proteins are selectively “loaded” into EVs. Also, the word “indicates” should be substituted with “suggests”.

Line 410: “It can be assumed”. Suggest changing this to “we hypothesize”

Line 441: “The latter is supported by EVsMOD proteomic data...” Perhaps the authors could also discuss how the EV RNA data could support the idea of EVsMOD have an alternative function.

Discussion: The authors propose a novel model whereby EV's represent a form of nutrient storage for subsequent utilization by the bacteria when nutrients become limited. While this is a novel proposal the discussion of this hypothesis is underdeveloped and should be expanded. What are the evolutionary advantages of extracellular storage of nutrients compared to intracellular storage? If EVs serve as a nutrient reservoir, would they not also be accessible to other bacterial species? What are the potential implications for this (in a microbiome setting). Couldn't EV's diffuse away from/be washed away from the bacterial cells (particularly during infection)? Can the authors clarify if they feel nutrient storage is the primary function of EVs or just one of a number of functions. I think these points and many others should be incorporated into the discussion.

Line 488: “For the production of EVs, bacteria were grown in the respective media at 30 °C”. Were both *B. cereus* and *S. aureus* strains grown at 30°C? Can the authors explain the reason behind their decision to grow the bacteria at 30 °C? Is this the most commonly used temperature for *B. cereus* growth? How would EV membrane lipid composition be impacted if *B. cereus* was grown at 37°C compared to 30°C? Growth of *S. aureus* is typically performed at 37 °C and EV composition in *S. aureus* has been shown to be temperature dependent (as is membrane lipid composition), therefore some explanation regarding the decision why this temperature was chosen should be given.

None of the data sets generated (proteomic/transcriptomic/lipidomic) have been deposited at an online database and made available to reviewers. This should have been done prior to manuscript submission. This data must be made available.

Reviewer #2

(Remarks to the Author)

In this paper, the authors examine the degradation of MVs and their role as a nutrient source. The turnover of MVs is a highly important topic particularly in terms of understanding their mobility and function but remains largely unexplored. I believe the aim of this study offers new insights into MV stability. However, several critical points need clarification, and the manuscript may be misleading due to insufficient discussion of key information. Notably, based on the current experiments and data, it is difficult to determine whether the authors are analyzing free sphingolipids or yeast EV originating from LB medium, or true bacterial MVs that have incorporated sphingolipids. If the strain does not produce sphingolipids and considering the data

showing that the EVs in LB medium mainly consist of sphingolipids, the most logical conclusion is that the authors are examining EVs derived from the medium (from yeast), rather than bacterial MVs. Below are specific comments.

Major comments

1. The authors mention very briefly in the discussion (line 444) that this bacterium does not produce sphingolipids. This information is very important but is hidden in the manuscript and should be clearly stated at the beginning of the manuscript. Whether the strain the authors used produces sphingolipids or not in the used condition will directly impact the interpretation of the entire dataset.
2. Do the authors have any evidence that the strain they used do not produce sphingolipids? To my knowledge, *B. cereus* in general do not produce sphingolipids, but I am unfamiliar of the strain the authors used. Whether this strain produce sphingolipids or not is a critical point which is not examined. Adding lipidomic data of the cells in table 1 will be useful in discussing the differences of the lipidomes of cells and MVs, and the medium if necessary to estimate which lipids comes from the medium. In addition, As commented further below, it is essential to purify the MVs instead of using the crude precipitation sample in analyzing the MV composition.
3. Most importantly, given that the sphingolipids are likely coming from the LB medium, it is critical to purify MVs with density gradient ultracentrifugation. Without purification, the authors may be analyzing LB-derived free sphingolipid or yeast EVs rather than bacterial MVs. Furthermore, it is critical to show that the sphingolipid is truly associated with bacterial MVs to reach the conclusion of this manuscript. Since over 70% of the lipidome in MVs(LB) consist of sphingolipids, it is possible that the majority of EVs in this condition is yeast derived EVs with bacterial MVs being a minor population.
4. The authors should analyze LB medium without bacterial inoculation and examine the "MV fraction" from this control. This would help determine whether the observed effects are due to bacterial MVs or to yeast extract–derived EVs or free sphingolipids present in the medium.
5. If bacterial MVs are involved and the strain do not produce sphingolipids, the overall data is suggesting that the MV lipid composition can be altered by extracellular lipid but is not well investigated. The authors should examine whether sphingolipid can be incorporated in MVs (or if bacterial MVs become associated with yeast EVs).
6. Related to the point above the authors indicate in Fig. 7 and elsewhere that the MVs derived from LB consist of sphingolipids and other lipids. If the EV mainly consist of sphingolipids, it could be degraded but if sphingolipid is not a major component, I guess the particle may remain intact. I fully understand it is challenging to see the lipid composition in a single particle but the heterogeneity of EVs should be taken into consideration.
7. It is unclear how the MVs were quantified, for example in fig. 1b. were the MVs first isolated and then counted ? If only crude precipitation was used without further purification, other particles such as flagella could be present and might be included in the NTA counts. This should be clarified.
8. It is better to show TEM image of a wider view rather than showing one image of a particle. From the image provided it is hard to tell if they are EV or not. Better images might be obtained if ultrathin sections are not used.

Version 1:

Reviewer comments:

Reviewer #2

(Remarks to the Author)

The authors did an excellent job addressing the concerns. The revised manuscript is clearer, and the additional experiments demonstrate how incorporation of sphingolipids can alter the fate of MVs, which is important for understanding MV function and turnover.

I do not have any further comments.

REVIEWER COMMENTS

Reviewer #1 (Remarks to the Author):

In this study the authors present novel data demonstrating that bacteria can utilize extracellular vesicles as a nutrition source under certain conditions. The ability to utilize EVs as a nutrition source is dependent on two factors (i) the lipid composition of the EV membrane and (ii) the ability of the bacteria to secrete an enzyme that degrades the EV, thus releasing its contents. Overall the experiments are well performed and appropriate controls are included. There are a number of places where additional controls would have increased rigor of the study and I have outlined those below. The idea that EVs may function as a nutritional storage system is novel and the data presented support the hypothesis that this may be possible under certain circumstances. Since this is a novel hypothesis I feel it's important the authors provide a more detailed discussion about how this impacts our knowledge of bacterial communities and the role of EVs (see comment below).

Author response: We thank the reviewer for the positive feedback. Further, we appreciate the constructive comments, which have helped improve our manuscript. As suggested, additional controls have been included, and a paragraph has been added to the discussion that addresses the potential impacts of the newly discovered function of bacterial EVs as a nutritional resource on bacterial communities and on the role of EVs. Responses to all comments are indicated in blue, and changes to the manuscript are highlighted in yellow.

Please refer to the section below for our point-by-point response.

The title of the manuscript is somewhat vague and cryptic. Perhaps consider rephrasing “provides new clues on bacterial vesicle function” with something like “suggests a role for EVs in nutrient storage/recycling”?

Author response: We thank the reviewer for this suggestion and changed the title accordingly.

Lines 1-2: Sphingomyelinase-mediated extracellular vesicle degradation suggests a role of EVs in nutrient recycling.

Line 107: “...were found to be larger than EVs_{MOD}”. Please be specific. How much larger were they?

Author response: We revised the following section on line 107 and added details about the EV size.

Lines 104-107: Nanoparticle tracking analysis (NTA; Supplementary Fig. 2g) was used to assess the size of EVs. EVs_{LB} (NTA: 195.3 nm) were found to be larger than EVs_{MOD} (NTA: 142.3 nm), resulting in a size difference of 53.0 nm, which was validated by isolating EVs from three additional pathogenic *B. cereus* strains (Supplementary Fig. 2g).

The results shown in supplemental fig 2d and e are important to the study and should be moved to the main manuscript.

Author response: We appreciate the valuable feedback. As requested, we incorporated the Supplementary Fig. 2d and 2e into Fig. 1 of the main manuscript. Accordingly, these figures are now Fig. 1g and Fig. 1h. We also updated the numbering of the following figures and the legends for Fig. 1 and for Supplementary Fig. 2.

Line 144: “The dynamics of bacterial EV production were monitored...”. Can the authors clarify if EV production, EV degradation, or both was measured in this assay?

Author response: To clarify this point, we have revised this statement on *B. cereus* as well as the respective statement on *S. aureus* in the manuscript as follows:

Lines 126-128: The number of *B. cereus* EVs was monitored in bacterial cultures grown in parallel in hfRPMI and unconditioned RPMI.

Lines 134-137: To test whether EV degradation is specific for *B. cereus* EVs, or whether it also occurs in other bacteria, *Staphylococcus aureus* Newman was cultured in different growth media (LB, hfRPMI, MOD, and RPMI) at 30 °C (Supplementary Fig. 3b). Subsequently, EV degradation was assessed by quantifying the number of EVs remaining in the conditioned media from each growth condition.

Line 178: “To assess whether EV degradation manifests in the particular proteomes...” Please rephrase/clarify the meaning here, it is unclear.

Author response: We apologise for this unclear phrasing and revised the manuscript as follows:

Lines 165-168: To investigate the potential impact of EV degradation on the *B. cereus* EV proteome, we performed an untargeted LC-MS/MS proteomic analysis of EV_{LB} and EV_{MOD} harvested from bacterial cultures after 9 h and 15 h.

Line 165: When FTIR spectroscopy is first introduced in the manuscript it would be useful to include a sentence or two outlining what this technique is used for to assist readers who are not familiar with the approach.

Author response: Done as suggested.

The following information on FTIR was included:

Lines 150-154: FTIR is a high-resolution vibrational spectroscopic technique that provides detailed insight into the molecular composition of biological samples^{25,26}. Based on stretching and bending

vibrations of functional groups of the molecular constituents, encompassing lipids, proteins, polysaccharides and nucleic acids, global molecular fingerprints can be generated^{14, 27}.

Line 173: Fig 1g should be Fig 1i

Author response: We thank the reviewer for pointing out the issue with the subfigure labelling. As Fig. 1 was modified in response to the aforementioned comment, the subfigure lettering has also been updated accordingly.

Line 178: “To assess whether EV degradation manifests in the particular proteomes...” Please rephrase this sentence as the meaning is unclear.

Author response: We rephrased the sentence as follows:

Lines 165-168: To investigate the potential impact of EV degradation on the *B. cereus* EV proteome, we performed an untargeted LC-MS/MS proteomic analysis of EV_{SLB} and EV_{SMOD} harvested from bacterial cultures after 9 h and 15 h.

Line 188: “The volcano plot visualises exclusive enrichment of proteins at 9 h in EVSLB (Supplementary Fig. 4a-d), further supporting the hypothesis that EVsLB degrade over time (Supplementary Fig. 4a)”. I don’t see how the data presented in volcano plots supports these statements. The data in Supplementary Fig. 4a seems to indicate a roughly equivalent number of proteins are more abundant at 15 hours and 9 hours. Can the authors clarify?

Author response: We agree that the wording may have led to an overestimation of the volcano diagrams in relation to EV_{SLB}. We changed the text accordingly.

It now reads as follows:

Lines 177-179: The high number of differentially regulated proteins (DEPs) observed between the EV_{SLB} 9 h and EV_{SLB} 15 h proteomes pinpoints the impact of EV degradation on the EV proteome (Supplementary Fig. 4a).

Please explain the origin of the data in figure 2D in more detail. The figure legend says it was constructed using differentially regulated proteins from the comparison presented in supplementary Figure 4 which helps, however only minimal details are given in the manuscript text. Please also explain what is meant by “The enrichment of non-redundant pathways” on line 192

Author response: We appreciate the reviewer's feedback and have accordingly expanded the data origin details for Fig. 2d in the main manuscript text and clarified the meaning of the sentence in line 192.

Lines 181-188: The DEPs [Differentially regulated proteins] in *B. cereus* EV_{LB} compared to *B. cereus* EV_{MOD}, determined at two different timepoints (9 h and 15 h; Supplementary Fig. 4c, d), were subjected to Gene Ontology (GO) enrichment analysis. This resulted in the identification of enriched pathways for each comparison. Because the GO hierarchy contains many redundant or overlapping terms, a redundancy-reduction step was applied to obtain a non-redundant set of pathways. Redundant GO terms were identified using a semantic similarity-based clustering implemented in the Python workflow with GOATOOLS⁵. The enriched pathways, cleared of duplicate and redundant terms, were visualised using a Sankey Diagram (Fig. 2d).

In several places throughout the manuscript the term “plain LB” is used. Does this refer to sterile unused LB? If so please clarify and use the term “sterile LB” instead.

Author response: Done as requested.

Plain LB changed to sterile LB, throughout the manuscript and in the supplement. **Lines: 108, 231, 546, 553 and 1096**

An important control is missing from Fig 4B. The SMase inhibitor (imipramime) should have been added along with exogenous SMase to show specific inhibition of SMase activity restores EV degradation.

Author response: We agree that this control is important and have performed the experiment as requested (**see new Supplementary Fig. 7f**). While the addition of SMase alone resulted in degradation of EVs, the simultaneous addition of Imipramine effectively inhibited SMase-mediated EV degradation.

The following paragraph was incorporated into the results section:

Lines 251-253: To confirm the specificity of Imipramine under our experimental conditions, we spiked sterile LB media with EV_{LB} and SMase. The addition of Imipramine blocked EV degradation, demonstrating that Imipramine specifically inhibits SMase activity (Supplementary Fig. 7f).

Line 257: “Various protease inhibitors tested in parallel did not inhibit this temperature-dependent, enzyme-driven process as effectively as the SMase inhibitor Imipramine.” This statement is misleading and is not supported by the results in supplemental figure 6b. Results in supplemental fig 6b show that addition of proteases did significantly inhibit EV degradation and in some cases (aprotinin) this inhibition appears to be comparable to or more than that observed with SMase inhibitor (imipramime). This needs to be explained/discussed in more detail. The iss

Author response: We apologise for this misleading statement and have included additional experimental data for clarification.

In the new experiments (**see new Supplementary Fig. 7e**), we used an SMase activity assay to evaluate the effects of Imipramine, Aprotinin, EDTA and PMSF on *B. cereus* SMase activity. Imipramine showed a strong inhibition of SMase, confirming its specific inhibition of *B. cereus* SMase. Furthermore, Aprotinin inhibited SMase activity to a lesser but still significant extent, while PMSF did not show a significant effect in the SMase activity assay. Aprotinin and PMSF are both serine protease inhibitors; however, their mechanisms of action differ, which may account for their divergent effects on SMase activity and SMase-mediated EV degradation. As it has been reported previously that the activation of SMase depends on certain serine proteases (DOI: 10.1096/fasebj.11.8.9240970), it is tempting to speculate that the inhibitory effect of Aprotinin is linked to the SMase activation cascade. However, further research will be necessary to decipher the mode of action of Aprotinin on SMase activity. The slight inhibition of EV degradation by PMSF is likely attributable to indirect effects rather than specific SMase inhibition. Consistent with the weak effect on EV degradation (Supplementary Fig. 7d), EDTA showed only weak inhibition in the SMase activity assay (**see new Supplementary Fig. 7e**). This weak inhibitory effect is likely to be associated with the chelation of divalent cations by EDTA (DOI: 10.1002/mrc.5009), given that SMase activity is dependent on divalent cations such as Co^{2+} and Mg^{2+} (DOI 10.1074/jbc.M601089200). Overall, these results indicate that understanding the interactions between protease inhibitors and *B. cereus* SMase will require a more systematic, mechanistic approach, which is clearly beyond the scope of the current study.

The following statement is now included in the manuscript:

Lines 238-250: Except for Aprotinin, the protease inhibitors tested in parallel did not inhibit this temperature-dependent, enzyme-driven process as effectively as the SMase inhibitor Imipramine (Supplementary Fig. 7b-d). Next, to assess whether protease inhibitors affect *B. cereus* SMase activity, we measured SMase activity with a fluorometric enzyme activity assay in the presence of Imipramine, Aprotinin, EDTA and PMSF, respectively (Supplementary Fig. 7e). Imipramine strongly inhibited SMase activity, confirming its direct inhibition of *B. cereus* SMase. Furthermore, Aprotinin inhibited SMase activity to a lesser but significant extent, while PMSF did not show a significant effect in the SMase activity assay. Aprotinin and PMSF are both serine protease inhibitors; however, their mechanisms of action differ, which may account for their divergent effects on SMase activity and SMase-mediated EV degradation. Consistent with the weak effect on EV degradation (Supplementary Fig. 7d), EDTA showed only weak inhibition in the SMase activity assay (Supplementary Fig. 7e). Given that SMase activity is dependent on divalent cations such as Co^{2+} and Mg^{2+} ³¹, this weak inhibitory effect is likely to be associated with the chelation of divalent cations by EDTA³².

Fig 4 a, c, and e. The schematics for each experiment are actually somewhat confusing and don't really add to the figure. At the bottom of each diagram the symbols used are depicted presumably as a "key" for the figure. However, at first it appears that they are part of the schematic figure itself. Perhaps enclosing them

in a box would make this more obvious and help with clarity of the figure. Fig 4e in particular is not helpful as it doesn't outline the experiment well.

Author response: We have modified the figure accordingly.

As suggested, the symbols used were enclosed in a box and Fig. 4e was reworked. Furthermore, Supplementary Fig. 6f was moved from the supplement to the main figure (see following comment) and was integrated into Fig. 4 (now Fig. 4g).

The result in supplemental fig 6f is quite important to the overall study and should perhaps be moved to the main manuscript.

Author response: Done as requested.

We moved Supplementary Fig. 6f to the main manuscript and integrated it into Fig. 4 (now Fig. 4e, g). We have also included a corresponding experimental schematic (see comment above).

Fig 5a and b. Could the authors provide this data as a plot of growth (OD600 Vs time) as supplemental data as this is the format most researchers are familiar with for bacterial growth.

Author response: Done as requested.

We have included the corresponding OD600 growth curve in the supplement (Supplementary Fig. 9a).

Line 297: "we screened the proteome data..." What proteome data are they referring to? Is it the data generated in this study (depicted in Fig 2?). If so this needs to be clearly stated. Also, how was the data screened? More details need to be provided.

Author response: We have reworked the section and included more details on the data generation and screening. The proteomics dataset generated in this study from *B. cereus* EVs (PXD065751) was uploaded to ShinyGo (<https://bioinformatics.sdstate.edu/go/>) and assigned to gene ontology (GO) terms. The assigned GO terms were searched for the parent term 'response to stimuli' (GO:0050896) and its subordinate terms (descendant terms in the GO hierarchy structure, *aka* child terms).

The following statement was added:

Lines 297-304: Therefore, we examined the proteomics dataset generated in this study from *B. cereus* EV (Supplementary Table 1; EV_{LB} 9h, EV_{LB} 15h, EV_{MOD} 9h, EV_{MOD} 15h) for proteins associated with environmental adaptation (Fig. 2). To obtain a comprehensive profile of the stress response proteome, the GO annotations for biological processes were first retrieved for all identified *B. cereus* EV proteins. Subsequently, the proteins associated with the parent GO term 'response to stimuli'

(GO:0050896) and its child terms (descendant terms in the GO hierarchy structure) were selected. The intensities of the selected proteins were visualised in the heat-map depicted in Fig. 5c.

Line 299: “(child-terms of the ‘response to’ GO-term Supplementary Table 4)”. What is meant by this statement? The meaning is unclear.

Author response: Done.

This section has been reworded to clarify and now reads as follows (see also comment to Line 297 above):

Lines 300-304: To obtain a comprehensive profile of the stress response proteome, the GO annotations for biological processes were first retrieved for all identified *B. cereus* EV proteins. Subsequently, the proteins associated with the parent GO term ‘response to stimuli’ (GO:0050896) and its child terms (descendant terms in the GO hierarchy structure) were selected. The intensities of the selected proteins were visualised in the heat-map depicted in Fig. 5c.

Line 309: “the exclusive enrichment of Dps1, Dps2 and the putative carbon starvation protein A in EVs_{LB} indicates that bacteria cultivated in LB experience nutritional stress.” Can the authors clarify whether they believe EV cargo proteins are present simply because that are simultaneously abundant in the cytoplasm of the originating cell, or if they believe proteins are selectively “loaded” into EVs.

Author response: Although proteins such as Dps1, Dps2 and CstA were enriched in EVs_{LB}, we do not interpret this as evidence for an active or selective loading mechanism. Instead, their presence in EVs_{LB} likely reflects the protein composition of the parental cell (protein abundance and spatial organisation) at the moment the EVs were formed. Therefore, we believe that enrichment of these nutritional stress-associated proteins in EVs_{LB} is due to passive loading. Nevertheless, we cannot exclude the possibility of a hitherto unknown active, selective loading mechanism.

Also, the word “indicates” should be substituted with “suggests”.

Author response: Done as requested. **(Line 315)**

Line 410: “It can be assumed”. Suggest changing this to “we hypothesize”

Author response: Changed as suggested. **(Line 376)**

Line 441: “The latter is supported by EVs_{MOD} proteomic data...” Perhaps the authors could also discuss how the EV RNA data could support the idea of EV_{MOD} have an alternative function.

Author response: We thank the reviewer for this suggestion and have included the following paragraph in the discussion:

Line 395-404: Thus, it is tempting to speculate that EVs_{MOD} have different functions, such as mediating stress signals. The latter is supported by *B. cereus* EVs_{MOD} proteomic data, which suggests that MOD-grown bacteria face translational and oxidative stress. The stress response reflected by the EVs is further supported by the increased abundance of ribosomal proteins and rRNA in EVs_{MOD}. Since bacteria lack strict compartmentalisation, they depend on a tightly regulated proteostasis to control protein production, folding, and transport. Exposure of bacteria to stressful and harsh environments results in an imbalance between transcription and translation⁴⁴⁻⁴⁶. Consequently, it is tempting to speculate that the loading of excess rRNA into EVs_{MOD} enables the bacterial cell to fine-tune its translational landscape. However, further research is necessary to determine the potential role of EVs in translational regulation in bacteria.

Discussion: The authors propose a novel model whereby EV's represent a form of nutrient storage for subsequent utilization by the bacteria when nutrients become limited. While this is a novel proposal the discussion of this hypothesis is underdeveloped and should be expanded. What are the evolutionary advantages of extracellular storage of nutrients compared to intracellular storage? If EVs serve as a nutrient reservoir, would they not also be accessible to other bacterial species? What are the potential implications for this (in a microbiome setting). Couldn't EV's diffuse away from/be washed away from the bacterial cells (particularly during infection)? Can the authors clarify if they feel nutrient storage is the primary function of EVs or just one of a number of functions. I think these points and many others should be incorporated into the discussion.

Author response: We thank the reviewer for the remarks on our discussion. We do not believe that nutrient storage is the primary or exclusive function of EVs, but rather one of several critical functional roles, as e.g., EVs_{MOD} are also secreted but do not play a significant role in EV-mediated nutrient recycling. We have expanded the discussion section, taking into account the potential effects of selective EV degradation and elaborating on the benefits and trade-offs of extracellular or intracellular nutrient storage. We do hope that our work, particularly the new aspects included in the extended discussion, inspires the scientific community to explore further the role of EVs in bacterial communities and bacterial-environmental interactions.

The newly introduced section reads as follows:

Lines 419-441: From an evolutionary perspective, extracellular nutrient storage in EVs provides a mechanism for bacterial populations to buffer transient nutrient limitation, extending beyond the

confines of the individual cell. By this mechanism, bacterial EVs can support not only the producing cell but also neighbouring bacteria sharing the same environment, thereby promoting collective survival under nutrient-limited conditions. Such EVs may act as 'selectively accessible packages', where only SMase-secreting bacteria, such as *Clostridium perfringens*, *Listeria monocytogenes*, and *Staphylococcus epidermidis*⁴⁰, can access the stored nutrients, adding a layer of specificity within microbial communities. However, further research will be required to explore the extent to which this extracellular nutrient storage can be used as a public good by the entire bacterial community or whether it confers a selective advantage for the producing bacteria or particular members of the bacterial community. Extracellular storage inevitably carries the disadvantage of potential loss through diffusion or environmental clearance. Yet this diffusion can also be advantageous, e.g. by facilitating host-pathogen interactions by systemic dissemination of EVs^{11,15} or priming distant niches, similar to mechanisms described for EVs in cancer metastasis⁵⁶. By contrast, intracellular storage allows rapid access to nutrients but restricts benefits to single cells, highlighting a trade-off between single-cell and community-level resource management strategies.

SMase-mediated EV degradation also reframes current concepts of EV functionalities. EVs protect, package, and deliver a wide range of bioactive molecules^{14,37,57,58}. By actively modulating EV turnover, SMase impacts EV stability and specificity, and may also affect quorum sensing³⁷, toxin delivery¹⁴, nutrient acquisition⁵⁸ and DNA transfer⁵⁹. Moreover, SMase-mediated EV degradation likely impairs EVs' ability to act as decoys against phages and antibiotics^{60,61}. Deciphering the precise contribution of SMase-mediated EV degradation to the functional repertoire of EVs will be crucial to understanding how EV functionality is integrated into the regulatory networks that shape bacterial communities.

Line 488: "For the production of EVs, bacteria were grown in the respective media at 30 °C". Were both B. cereus and S. aureus strains grown at 30°C? Can the authors explain the reason behind their decision to grow the bacteria at 30 °C? Is this the most commonly used temperature for B. cereus growth? How would EV membrane lipid composition be impacted if B. cereus was grown at 37°C compared to 30 °C? Growth of S. aureus is typically performed at 37 °C and EV composition in S. aureus has been shown to be temperature dependent (as is membrane lipid composition), therefore some explanation regarding the decision why this temperature was chosen should be given.

Author response: Indeed, 30 °C is the temperature commonly used for *B. cereus* research and diagnostics, as now mentioned in the Methods section of the revised manuscript.

Lines 485-486: For the production of EVs, bacteria were grown in the respective media at 30 °C, the commonly used *B. cereus* growth temperature, and 120 rpm shaking as described previously⁷⁴.

To ensure comparability between the *B. cereus* and *S. aureus*, *S. aureus* was also cultivated at 30 °C. We agree that the EV membrane lipid composition might be temperature dependent. Therefore, we investigated the amount of sphingomyelin, the target of SMase, in *B. cereus* and *S. aureus* Newman EVs that were isolated from bacterial cultures grown at either 30 °C or 37 °C. This analysis revealed no significant difference in the sphingomyelin quantity between 30 °C and 37 °C in either *B. cereus* or *S. aureus* (Explanatory Fig. 1).

Explanatory Fig. 1: Sphingomyelin levels of *B. cereus* and *S. aureus* EVs_{LB} are similar between 30 °C and 37 °C. *B. cereus* and *S. aureus* were grown in LB at either 30 °C or 37 °C for 7 h, and EVs were harvested by differential ultracentrifugation. Subsequently, the sphingomyelin content was quantified using a commercially available fluorometric sphingomyelin quantification assay (as described in the Methods section).

None of the data sets generated (proteomic/transcriptomic/lipidomic) have been deposited at an online database and made available to reviewers. This should have been done prior to manuscript submission. This data must be made available.

Author response: We thank the reviewer for pointing this out and apologise for the inconvenience. All datasets are uploaded and accessible for the reviewers using the following key:

Proteomics: <https://proteomecentral.proteomexchange.org/cgi/GetDataset?ID=PXD065751>,
reviewer account username: reviewer_pxd065751@ebi.ac.uk, **password:** 2kGLi3g78kdH

Lipidomics: <https://www.ebi.ac.uk/metabolights/editor/MTBLS13260/files?reviewCode=0816091-2cde-4f0b-b970-3c2275427ec8>

Transcriptomics: <https://www.ebi.ac.uk/ena/browser/view/PRJEB94405>

Statements pertaining to data availability have been incorporated into the respective methods sections.

Lines 621-622: The proteomics data have been deposited to the ProteomeXchange Consortium with the following identifier PXD065751.

Lines 709-710: The lipidomics data have been deposited to MetaboLights repository with the study identifier MTBLS13260⁹⁸.

Lines 660-661: The transcriptomics data have been deposited to European Nucleotide Archive with the following accession number PRJEB94405.

In addition, we have included a data availability statement covering all three omics approaches at the end of the methods section, including the access information for the reviewing process.

Lines 769-779: Source Data are provided with this paper. The mass spectrometric raw files as well as the MaxQuant output files concerning proteomics have been deposited to the ProteomeXchange Consortium via the PRIDE repository and can be accessed using the identifier PXD065751 (<https://proteomecentral.proteomexchange.org/cgi/GetDataset?ID=PXD065751>), **reviewer account username: reviewer_pxd065751@ebi.ac.uk, password: 2kGLi3g78kdH**). The mass spectrometric raw files as well as the output files concerning lipidomics have been deposited to the MetaboLights repository with the study identifier MTBLS13260 (<https://www.ebi.ac.uk/metabolights/editor/MTBLS13260/files?reviewCode=00816091-2cde-4f0b-b970-3c2275427ec8>). The transcriptomic data is deposited at European Nucleotide Archive under the ENA Accession Number PRJEB94405 (<https://www.ebi.ac.uk/ena/browser/view/PRJEB94405>) and is available during the review process.

Reviewer #2 (Remarks to the Author):

In this paper, the authors examine the degradation of MVs and their role as a nutrient source. The turnover of MVs is a highly important topic particularly in terms of understanding their mobility and function but remains largely unexplored. I believe the aim of this study offers new insights into MV stability. However, several critical points need clarification, and the manuscript may be misleading due to insufficient discussion of key information. Notably, based on the current experiments and data, it is difficult to determine whether the authors are analyzing free sphingolipids or yeast EV originating from LB medium, or true bacterial MVs that have incorporated sphingolipids. If the strain does not produce sphingolipids and considering the data showing that the EVs in LB medium mainly consist of sphingolipids, the most logical conclusion is that the authors are examining EVs derived from the medium (from yeast), rather than bacterial MVs. Below are specific comments.

We thank the reviewer for the remarks and valuable suggestions. We especially appreciate that the research question addressed in this work is considered highly relevant.

Firstly, we would like to extend our apologies to the reviewer for not presenting the full kinetics of EVs from *B. cereus* cultures in Fig. 1b. As the primary aim of our research pertains to the degradation of EVs, the initial presentation of data in Fig 1b was focused on the late growth phases of *B. cereus*. From the reviewer's inquiries concerning the potential implications of EV yeast extract-derived EVs from the LB medium used for bacterial cultivation, we assume that not showing data from the early growth phase in Fig 1b has led to a misunderstanding regarding the origin of EVs, for which we apologise.

EV yields from bacterial cultures at 0 h (directly after inoculation) and after 3 h of cultivation have now been included in Fig. 1b of our revised manuscript. Additionally, as suggested by the reviewer, we performed NTA analyses of sterile growth media controls and incorporated the results into Fig. 1b. To this end, sterile LB and MOD were subjected to ultracentrifugation and potentially pelleted particles were analysed using NTA. No EV particles were detected, neither in the yeast-extract-containing LB nor in the defined mineral medium MOD. The revised Fig. 1b (depicted below for information together with the experimental scheme Fig. 1a) shows the specific enrichment of bacterial EVs during the early growth phase and the absence of potential contaminating EVs originating from the sterile media used for bacterial cultivation (LB, MOD). Specifically, a 70-fold increase in EV yields from *B. cereus* cultures was observed between 3 h and 6 h in both LB and MOD. Following a 6 h incubation period, a substantial decline in EV yields was observed for *B. cereus* LB cultures, while a concomitant increase in EVs was evident for *B. cereus* MOD cultures. These results show that the observed effects in our experiments are attributable to bacterial EVs, not to potential contaminating yeast-extract EVs or free sphingolipids in the media.

REDACTED

Details of revised Fig. 1a, b. EV production of *B. cereus*. a To investigate EV secretion in nutrient-rich and nutrient-scarce cultivation environments, EVs of *B. cereus* F4810/72 were harvested 0, 3, 6, 9, 12 and 15 h post-inoculation, and subjected to differential (ultra)centrifugation to obtain EVs. In addition, media controls were prepared by subjecting sterile media to ultracentrifugation. b EVs were isolated from bacterial cultures via ultracentrifugation. The pelleted EVs were reconstituted in 30 μ l PBS and EV yields were determined by nanoparticle tracking analysis (NTA) for each time point as indicated on the x-axis (n=4 biological replicates). In addition, sterile media controls were subjected to the same EV isolation protocol described above and NTA was performed (n=3 independent media preparations). Abbreviations: EV_{LB}: EVs isolated from *B. cereus* cultures grown in LB; EV_{MOD}: EVs isolated from *B. cereus* cultures grown in MOD; nd: not detected.

These findings, as well as the results of the additional experiments suggested by the reviewer (see response to claims 3, 4, 8), conclusively demonstrate that bacterial EVs, not EVs derived from the medium (from yeast), are examined throughout our work. This conclusion is further supported by our EV proteome data, which demonstrate that 99.84 % of proteins are assigned to *B. cereus*, while 0.16 % are assigned to yeast (see Claim 3.2, Explanatory Fig. 3).

We hope this explanation, along with the results from the additional experiments, has clarified any open questions. In summary, we believe that our manuscript has benefited from the inclusion of the proposed additional controls and experiments.

Please find below our detailed response to each comment. Responses to all comments are indicated in blue, and changes to the manuscript are highlighted in yellow.

1. The authors mention very briefly in the discussion (line 444) that this bacterium does not produce sphingolipids. This information is very important but is hidden in the manuscript and should be clearly stated at the beginning of the manuscript. Whether the strain the authors used produces sphingolipids or not in the used condition will directly impact the interpretation of the entire dataset.

Author response: We apologise for not clearly stating this information at the beginning of the manuscript.

The following statement is now included in the introduction:

Lines 55-61: Although progress has been made in understanding the functions of bacterial EVs, the role of the bacterial EV lipid bilayer remains poorly understood. For instance, sphingolipids are

repeatedly reported as major constituents of bacterial EV lipid bilayers¹⁸⁻²¹. Yet, the majority of bacterial species, including *B. cereus*, are unable to synthesise sphingolipids *de novo*^{22,23}. Nevertheless, they may be capable of scavenging sphingolipids from their environment²⁴. Despite consensus on the presence of sphingolipids in bacterial EVs, their functional role within bacterial EV membranes remains to be elucidated.

2. Do the authors have any evidence that the strain they used do not produce sphingolipids? To my knowledge, *B. cereus* in general do not produce sphingolipids, but I am unfamiliar of the strain the authors used. Whether this strain produce sphingolipids or not is a critical point which is not examined. Adding lipidomic data of the cells in table 1 will be useful in discussing the differences of the lipidomes of cells and MVs, and the medium if necessary to estimate which lipids comes from the medium. In addition, as commented further below, it is essential to purify the MVs instead of using the crude precipitation sample in analysing the MV composition.

2.1 Regarding the reviewer's question about sphingolipid production in *B. cereus*.

Author response: Correct. *B. cereus*, including the *B. cereus* strain used in this study, does not belong to the group of sphingolipid-producing bacteria. The lipidomics analysis of *B. cereus* EVs presented in this study demonstrates that sphingolipids are detected only in the EV lipidome when they are present in the growth environment (**see new Supplementary Fig. 5b**). Consistently, no sphingolipids were detected in EVs derived from bacteria grown in sphingolipid-free media. The inability to synthesise sphingolipids *de novo* is a general feature reported for many bacterial taxa, including members of the phylum Firmicutes (*aka* Bacillota) (DOI: 10.1073/pnas.1001501107). To our knowledge, *de novo* sphingolipid biosynthesis has been described primarily in eukaryotes and in a limited number of bacterial species, such as the phylum Bacteroidetes, Chlorobacteria, and subsets of Alpha- and Delta-Proteobacteria (DOI: 10.1073/pnas.1001501107, DOI: 10.3389/fmicb.2023.1289819, DOI: 0.1016/j.mib.2017.12.011, DOI: 10.1039/c8np00019k). That most bacteria, including *B. cereus*, are not able to produce sphingolipids is now mentioned in the introduction section of the revised MS (**Lines 55-61**; see also answer to claim 1 above).

2.2 Regarding the reviewer's comment about the origin of sphingolipids.

Author response: We agree with the reviewer that a comprehensive comparative lipidomic analysis of bacterial cells and culture media would be interesting; however, it would exceed the scope of the present study. The protocol used for EV lipid extraction for lipidomics is not suitable for bacterial cells. Thus, such a comparative analysis would require a different experimental procedure, which was beyond the scope of the current study.

To address the origin of sphingomyelin, a known bacterial SMase target (DOI: 10.1021/bi980915e; DOI: 10.1074/jbc.M601089200), we employed a targeted quantitative sphingomyelin assay. This assay confirmed that EV_{LB} are sphingomyelin-positive, whereas EV_{MOD} are sphingomyelin-negative (**see new Supplementary Fig. 5a**), consistent with the lipidomics data. We also attempted to quantify sphingomyelin in bacterial membranes; however, the sphingomyelin assay was not compatible with the organic solvent required for lipid extraction from bacterial cells. Due to the structure of the bacterial cell wall, the protocol used for EVs was not compatible with bacterial cells.

Thus, we used three additional approaches to address the origin of the sphingomyelins:

a) Spike-in experiments using fluorescently labelled sphingomyelin. *B. cereus* cultures were supplemented with a fluorescently labelled sphingomyelin (SM C11; 4 µM; C11 TopFluor™), extensively washed, and analysed by fluorescence microscopy, revealing membrane-associated fluorescence indicative of sphingomyelin uptake. Moreover, fluorescence measurements of isolated *B. cereus* EVs of those cultures confirmed the incorporation of SM C11 into the EVs (see for more details Claim 5; Explanatory Fig. 4a, b).

These experiments showed that *B. cereus* takes up sphingomyelins from the environment and incorporates them in EVs (for details, see answer to Claim 5 and the respective Explanatory Fig. 4a, b).

b) Quantification of sphingomyelin in media. To assess whether the sphingomyelin of bacterial EV_{LB} originates from sphingomyelin-scavenging of the EV-producing bacteria, we quantified the sphingomyelin content of the sterile cultivation media (LB, MOD). As expected, sterile LB was positive, whereas MOD was tested negatively (**see new Supplementary Fig. 5b**). To exclude that the sphingomyelins detected in our lipidomic analysis of EV_{LB} derived from any potentially pelletable sphingomyelin-positive particles in sterile LB, the sphingomyelin signal of sterile LB to sterile LB ultracentrifuged for 16 h, 180,000 x g (designated as LB*) was compared. No significant differences were detected between LB and LB* (**see new Supplementary Fig. 5b**). These results imply that a contamination of our bacterial EV_{LB} sample with the “LB-MV fraction” (see claim 4) can be excluded, fostering the notion that *B. cereus* can scavenge free sphingomyelins from the cultivation medium.

c) Comparison of the sphingomyelin content of density gradient-purified EVs and crude EVs. This experiment showed that crude EVs and density gradient-purified EVs contain equal amounts of sphingomyelin (**see new Supplementary Fig. 8b**), further supporting that sphingomyelin is an integral part of *B. cereus* EV_{LB} (see also response to claim 3).

A new paragraph on the new supplementary experiments was introduced into the results section

This paragraph reads as follows:

Lines 208-214: To further investigate the sphingomyelin content in *B. cereus* EVs, a quantitative sphingomyelin assay was performed, revealing a dose-dependent increase in sphingomyelins in EV_{LB},

whereas EV_{MOD} did not contain any detectable sphingomyelins (Supplementary Fig. 5a). To rule out contaminations from potentially pelletable particles in LB, we quantified sphingomyelin in sterile LB and sterile, ultracentrifuged LB (LB*^{*}; Supplementary Fig. 5b). Both media showed similar sphingomyelin levels, indicating the absence of sphingomyelin-positive pelletable particles (Supplementary Fig. 5b).

3. Most importantly, given that the sphingolipids are likely coming from the LB medium, it is critical to purify MVs with density gradient ultracentrifugation. Without purification, the authors may be analyzing LB-derived free sphingolipid or yeast EVs rather than bacterial MVs. Furthermore, it is critical to show that the sphingolipid is truly associated with bacterial MVs to reach the conclusion of this manuscript. Since over 70% of the lipidome in MVs(LB) consist of sphingolipids, it is possible that the majority of EVs in this condition is yeast derived EVs with bacterial MVs being a minor population.

3.1. Regarding the reviewer's suggestion to include results from MVs purified with density gradient ultracentrifugation:

In order to address the research question concerning the potential of bacterial EVs to serve as a nutrient reservoir, we are using crude EVs to simulate natural conditions. However, we agree with this reviewer that using EVs purified by density gradient ultracentrifugation would help to validate our findings and support our conclusions. Thus, as suggested, EVs were purified by means of density gradient ultracentrifugation (EV_{LB-DG}) and subjected to EV degradation experiments (see new Supplementary Fig. 8a). For these experiments, EV_{LB} and EV_{LB-DG} were spiked into sterile LB, sterile LB supplemented with external SMase, or sterile LB containing both external SMase and the SMase inhibitor Imipramine. The results from EV degradation experiments on EV_{LB-DG} mirrored results obtained from degradation experiments on crude EV preparations (EV_{LB}). Thus, it could be assumed that the SMase-mediated bacterial EV degradation described in this work is not caused by free sphingolipids present in the media.

In addition, EVs from bacteria grown in ultracentrifuged LB (LB*^{*}) were harvested by ultracentrifugation and further purified by density gradient ultracentrifugation (EV_{LB*^{*}-DG}) to assess a potential impact of LB media -derived particles on EV preparations. NTA analysis revealed a comparable size distribution for both EV_{LB-DG} and EV_{LB*^{*}-DG} (Explanatory Fig. 2).

The results from EV degradation experiments on EV_{LB*^{*}-DG} confirmed the results from the degradation experiments on crude EV preparations (EV_{LB} and EV_{LB*^{*}}). Irrespective of the method employed for EV preparation, EV degradation was observed only in the presence of SMase and was inhibited by the SMase inhibitor Imipramine. No significant differences were observed in the extent of SMase-mediated EV degradation between EV_{LB}, EV_{LB*^{*}}, EV_{LB-DG}, and EV_{LB*^{*}-DG} (new Supplementary Fig. 8a). Furthermore, we tested EVs purified by density gradient centrifugation for sphingomyelins (new

Supplementary Fig. 8b). Similar amounts of sphingomyelin were observed in $EV_{S_{LB-DG}}$, $EV_{S_{LB^*-DG}}$, $EV_{S_{LB}}$, and $EV_{S_{LB^*}}$, further supporting that sphingomyelin is an integral part of *B. cereus* $EV_{S_{LB}}$ (see also response to claim 2).

In summary, the supplementary experiments conducted with purified EVs corroborated our results using crude EVs and further substantiate our conclusions that bacterial $EV_{S_{LB}}$ can undergo SMase-mediated EV degradation (see **new Supplementary Fig. 8a, b**). We thank the reviewer for this suggestion.

Explanatory Fig. 2: NTA analysis of density gradient (DG) purified EVs originating from *B. cereus* grown in LB and ultracentrifuged LB (LB*). NTA was used to assess the size-distribution of DG purified EVs ($EV_{S_{LB-DG}}$ and $EV_{S_{LB^*-DG}}$; n=3).

A new paragraph on the new supplementary experiments using density gradient purified EVs was introduced into the results section (see **Lines 269-284**).

This paragraph reads as follows:

Lines 269-284: To exclude the possibility that co-purified factors may affect EV degradation, $EV_{S_{LB}}$ were further purified by density gradient ultracentrifugation ($EV_{S_{LB-DG}}$). In addition, to rule out the possibility that LB-derived particles interfere with the EV degradation experiments, EVs from *B. cereus* grown in ultracentrifuged LB (LB*) were harvested by ultracentrifugation ($EV_{S_{LB^*}}$) and further purified by density gradient ultracentrifugation ($EV_{S_{LB^*-DG}}$). Subsequently, $EV_{S_{LB-DG}}$ and $EV_{S_{LB^*-DG}}$ were used in spike-in experiments. Both $EV_{S_{LB-DG}}$ and $EV_{S_{LB^*-DG}}$ were spiked into sterile LB, sterile LB supplemented with external SMase, or sterile LB containing both external SMase and Imipramine, and the results were compared to those from spike-in experiments with $EV_{S_{LB}}$ and $EV_{S_{LB^*}}$ (Supplementary Fig. 8a). Irrespective of the method used for *B. cereus* EV preparation, EV degradation was observed exclusively in the presence of SMase and was inhibited by Imipramine. No significant differences were observed in the extent of SMase-mediated EV degradation between $EV_{S_{LB}}$, $EV_{S_{LB^*}}$, $EV_{S_{LB-DG}}$, and $EV_{S_{LB^*-DG}}$ (Supplementary Fig. 8a). Furthermore, EVs purified by density gradient centrifugation were tested positive for sphingomyelins (Supplementary Fig. 8b). Similar amounts of sphingomyelin were observed in $EV_{S_{LB-DG}}$, $EV_{S_{LB^*-DG}}$, $EV_{S_{LB}}$, and $EV_{S_{LB^*}}$. These findings demonstrate that density gradient-purified *B. cereus* EVs have the same sphingomyelin-positive lipid architecture and SMase susceptibility as crudely purified *B. cereus* EVs (Supplementary Fig. 8a, b).

3.2. Regarding the claim ‘it is possible that the majority of EVs in this condition [LB] is yeast derived EVs with bacterial MVs being a minor population’:

Based on our additional experiments and data analyses, we think that this can be ruled out for the following reasons:

1. **Results from electron microscopic analysis:** Sterile LB was analysed by electron microscopy. No EV structures were observed (see new Supplementary Fig. 2e), indicating that the yeast-extract-derived EVs from LB media are not a significant constituent of our EV_{LB} preparation.
2. **Results from NTA (see revised Fig. 1b):** We subjected sterile LB to ultracentrifugation and then analysed the potentially pelleted particles by NTA. No pelletable particles were detectable, demonstrating that yeast-extract-derived EVs from LB are not significant constituents of our bacterial EV_{LB}. Furthermore, immediately after inoculating LB with *B. cereus*, no EVs were detectable, but a marked increase in the EV yields occurred during the initial 6 h of incubation (see revised Fig 1b). These results show that *B. cereus* actively produces EVs in LB during the early growth phase.
3. **Results from our EV proteome analysis:** We assessed the presence of yeast proteins in our EV proteomics data by performing a database search against both the *Bacillus cereus* (proteome ID UP000002214) and the *Saccharomyces cerevisiae* (strain ATCC 204508/S288c; reference proteome ID UP000002311) protein database.

This analysis revealed a negligible background of yeast-proteins, with a mean contribution of 0.16 % across all replicates, while 99.84 % of the proteins were assigned to *B. cereus* (Explanatory Fig. 3). To statistically validate it, a paired t-test was performed on the log₁₀-transformed data, yielding a very significant p-value of $p = 9.93 \times 10^{-6}$. This p-value confirms that the yeast can be considered as background noise. It is mathematically improbable for a trace 0.16 % protein fraction to drive this high level of observed EV degradation and sphingolipid remodelling.

Explanatory Fig. 3: Analysis of the proteome data of EV_{LB}.

The contribution of yeast proteins to the EV-proteome data from EV_{LB} isolated from *B. cereus* cultures grown in LB, was investigated. Therefore, the mass spectrometric intensities (iBAQ values) of all proteins matching *S. cerevisiae* were summed and divided by the total protein intensity per sample to determine the contribution of yeast-derived proteins to the EV_{LB} proteome. The same procedure was performed on *B. cereus* proteins.

The text in the respective results section reads as follows:

Lines 95-104: In addition, we employed two complementary transmission electron microscopy (TEM) approaches: (i) imaging of ultrathin sections prepared from resin-embedded *B. cereus* EVs that were subsequently stained (Supplementary Fig. 2a, b), and (ii) the drop-on-grid method, in which *B. cereus* EVs were directly dropped onto TEM grids (Supplementary Fig. 2c, d). The TEM analyses confirmed that *B. cereus* EVs deriving from both media are lipid-bilayer bordered entities that are sphere-shaped. In the sterile media (LB, MOD) controls, no particle-like structures were detected by TEM (Supplementary Fig. 2e, f).

Lines 79-91: To investigate how nutrient complexity affects EV secretion in Gram-positive bacteria, we compared *B. cereus* EV yields from complex, nutrient-rich (LB) and defined, nutrient-scarce (MOD) media over time. The emetic *B. cereus* reference strain F4810/72, which served as a model organism, was grown either in LB or in MOD. EVs were isolated from bacterial cultures after 0 h, 3 h, 6 h, 9 h, 12 h and 15 h by differential centrifugation followed by ultracentrifugation, as outlined in figure 1a (Fig. 1a). The EVs isolated from *B. cereus* cultures were designated as EV_{LB} or EV_{MOD}, respectively. Immediately after inoculating LB and MOD with *B. cereus*, as well as in sterile media controls, no EVs were detectable (Fig. 1b). However, a marked increase in the *B. cereus* EV yields occurred during the initial 6 h of cultivation. Specifically, a 70-fold increase in EV yields was observed between 3 h and 6 h in both LB and MOD, resulting in comparable EV yields of EV_{LB} and EV_{MOD} at 6 h. Unexpectedly, EV yields decreased significantly after 9 h in LB but remained stable in MOD (Fig. 1b). This decrease might either reflect a different stability of EV_{LB} and EV_{MOD} or might be linked to distinctive factors actively degrading EV_{LB}.

4. The authors should analyze LB medium without bacterial inoculation and examine the "MV fraction" from this control. This would help determine whether the observed effects are due to bacterial MVs or to yeast extract-derived EVs, or free sphingolipids present in the medium.

Author response: We thank the reviewer for suggesting the incorporation of this negative control. As suggested, we have analysed the "MV fraction" in LB medium. To do so, we subjected sterile LB to ultracentrifugation and then analysed the potentially pelleted particles ("MV fraction") by NTA. No pelletable particles were detected in LB medium by our protocol (see also response claim 3 above). The results of the ultracentrifuged "MV fraction" control from sterile LB and MOD media are now included in the **revised Fig. 1b** and the results section (**Lines 84-88**).

To further validate the absence of EVs stemming from LB, TEM images of LB were obtained using the drop-on-grid method (**new Supplementary Fig. 2e**). No EV-like structures were observed, confirming the absence of yeast extract-derived EVs in LB medium used in our study.

As a growth-promoting effect in the later growth phase of *B. cereus* cultures in LB is observed only in the presence of EV_{LB} (see Fig. 6c), it could be assumed that the SMase-mediated bacterial EVs degradation and role of EVs in nutrient recycling described in this work is not caused by free sphingolipids present in the media.

In summary, these results confirm that the observed effects in our experiments are attributable to bacterial EVs, not to components of the LB medium.

The respective sections in the manuscript now read as follows:

Lines 84-88: Immediately after inoculating LB and MOD with *B. cereus*, as well as in sterile media controls, no EVs were detectable (Fig. 1b). However, a marked increase in the EV yields occurred during the initial 6 h of incubation. Specifically, a 70-fold increase in EV yields was observed between 3 h and 6 h in both LB and MOD, resulting in comparable EV yields of EV_{LB} and EV_{MOD} at 6 h.

Lines 95-104: In addition, we employed two complementary transmission electron microscopy (TEM) approaches: (i) imaging of ultrathin sections prepared from resin-embedded *B. cereus* EVs that were subsequently stained (Supplementary Fig. 2a, b), and (ii) the drop-on-grid method, in which *B. cereus* EVs were directly dropped onto TEM grids (Supplementary Fig. 2c, d). The TEM analyses confirmed that *B. cereus* EVs deriving from both media are lipid-bilayer bordered entities that are sphere-shaped. In the sterile media (LB, MOD) controls, no particle-like structures were detected by TEM (Supplementary Fig. 2e, f).

5. If bacterial MVs are involved and the strain do not produce sphingolipids, the overall data is suggesting that the MV lipid composition can be altered by extracellular lipid but is not well investigated. The authors should examine whether sphingolipid can be incorporated in MVs (or if bacterial MVs become associated with yeast EVs).

Author response: We thank the reviewer for this insightful comment. We concur that the influence of extracellular lipids on bacterial EV lipid composition remains poorly explored, and our study provides initial evidence that nutrient complexity shapes EV lipid architecture. Since *B. cereus* cannot synthesise sphingolipids, our findings suggest that medium-derived sphingomyelin can be taken up by *B. cereus* and subsequently incorporated into EV membranes.

To further experimentally support this idea, we grew *B. cereus* cultures in LB supplemented with fluorescently labelled sphingomyelin (SM C11; C11 TopFluor™ Sphingomyelin). After 6 h of cultivation, we isolated EVs via differential ultracentrifugation. Fluorescence microscopy revealed membrane-associated fluorescence in *B. cereus*, indicating uptake and incorporation of the exogenous supplemented SM C11 in *B. cereus* cell membranes (Explanatory Fig. 4a). Moreover, fluorescence measurements of the isolated *B. cereus* EVs (*B. cereus*^{+SM C11} EVs) showed that SM C11 is indeed incorporated in *B. cereus* EVs (see for details Explanatory Fig. 4b). The signal of the controls (EVs from

unlabelled *B. cereus* cultures (*B. cereus*^{-SM C11} EVs), sterile LB spiked with SM C11 (LB^{+SM C11} Control)) did not exceed the background signal obtained for sterile LB (LB^{-SM C11} Control).

Explanatory Fig. 4. SpHINGOMYELIN presented in the culture media are scavenged by *B. cereus* and incorporated into *B. cereus* EVs. **a** Bacteria were grown in LB, supplemented with 4 μ M of C11 TopFluor™ SpHINGOMYELIN (SM C11) solved in DMSO (upper panel). An equal amount of DMSO was added to the no-treatment control (lower panel). After 6 h of cultivation in the presence of SM C11 or DMSO, the bacterial cells were harvested by centrifugation and washed 3 times to remove any unbound dye. The final bacterial pellet was resuspended in PBS, applied on a cover slip, dried and covered using mounting media. Subsequently, cover slips were investigated by means of an inverted widefield microscope (Observer Z1, Zeiss) using phase contrast and an appropriate filter (excitation: 450-490 nm; emission: 515-565 nm). **b** EVs produced by *B. cereus* grown in LB plus SM C11 (*B. cereus*^{+SM C11} EVs) as well as in LB plus DMSO (*B. cereus*^{-SM C11} EVs) were isolated by ultracentrifugation, washed two times and the fluorescence intensities (excitation: 490 nm; emission: 530 nm) were analysed using a fluorescence microtiter plate reader (Spark® Multimode Microplate Reader, TECAN). Sterile media spiked with DMSO-solved SM (LB^{+SM C11} Control) was included to check for potential pelletable SM C11 aggregates. Sterile media supplemented with DMSO (LB^{-SM C11} Control) served as negative control.

Furthermore, as yeast EVs could not be detected or isolated in the LB medium (see also responses to claims 3 and 4 above, **revised Fig. 1b**, **newly added Supplementary Fig. 2e**), it can be assumed that bacterial EVs do not become associated with yeast-extract-derived EVs (originating from LB media) in our experimental setting. This notion is supported by our EV proteome analyses, which revealed that less than 0.2 % of identified proteins matched yeast proteins, confirming a minimal contribution of yeast-derived components to our EV preparations (see also response to claim 3.2 and Explanatory Fig. 3).

6. Related to the point above the authors indicate in Fig. 7 and elsewhere that the MVs derived from LB consist of sphingolipids and other lipids. If the EV mainly consist of sphingolipids, it could be degraded but if sphingolipid is not a major component, I guess the particle may remain intact. I fully understand it is

challenging to see the lipid composition in a single particle but the heterogeneity of EVs should be taken into consideration.

Author response: We agree with the reviewer that EV heterogeneity may affect EV degradation. So far, bacterial EV heterogeneity has been demonstrated mainly based on morphology (DOI: 10.3389/fmicb.2021.713669; DOI: 10.1128/Spectrum.01273-21). However, it is conceivable that the distribution of single lipid species among EVs is also heterogeneous. Such lipid heterogeneity may indeed influence susceptibility to SMase-mediated degradation. In our experiments, bulk measurements showed a steep decrease, but not complete loss, of EV_{LB} upon SMase treatment (Fig. 1b, h, j, Fig. 4b, d, f, g). Yet it remains to be elucidated whether this is attributable to SMase exhaustion or to the presence of a resistant subpopulation of EV_{LB}.

We agree that resolving and deciphering EV lipid heterogeneity will be pivotal for understanding EV biology in the forthcoming years, but this is clearly beyond the scope of the current study, which focuses on a novel role of EVs as a nutrient resource. Thus, the following statement was incorporated into the discussion:

Lines 445-459: Therefore, further insight into bacterial EV heterogeneity and the distribution of lipid species within EV populations is needed. While bacterial EV heterogeneity has so far been demonstrated mainly at the morphological level^{62,63}, it is plausible that different EV morphologies result from distinct EV lipid architectures, reflecting differences in membrane origin, curvature, or biogenesis pathways^{54,64,65}. Resolving EV lipid composition at the single-particle level will be crucial for determining whether all EV subtypes are equally susceptible to SMase-mediated degradation and for understanding how EV stability is regulated across heterogeneous populations. Emerging single-vesicle analytical approaches, such as direct stochastic optical reconstruction microscopy (dSTORM), Raman trapping analysis and single EV flow cytometry, allow analysing single EVs in eukaryotic systems⁶⁶⁻⁶⁸, but their applicability remains very limited for bacterial EVs due to the absence of specific marker proteins and the high diversity of bacterial species⁶⁹. Adapting and extending these technologies to bacteria will be essential to determine how bacterial communities utilise EV heterogeneity to fine-tune EV stability, cargo delivery and resource allocation, thereby offering further mechanistic insights into SMase-mediated EV degradation.

7. It is unclear how the MVs were quantified, for example in fig. 1b. were the MVs first isolated and then counted? If only crude precipitation was used without further purification, other particles such as flagella could be present and might be included in the NTA counts. This should be clarified.

Author response: We apologise for not stating this more clearly and have reworked the section. EVs were isolated by differential ultracentrifugation, and the EV yields were determined by NTA. A more detailed description is now given in the legend of Fig. 1b, which now reads as follows:

Lines 1049-1056: **a** To investigate EV secretion in nutrient-rich and nutrient-scarce cultivation environments, EVs of *B. cereus* F4810/72 were harvested 0, 3, 6, 9, 12 and 15 h post-inoculation – as depicted in the experimental scheme – and subjected to differential (ultra)centrifugation to obtain EVs. Media controls were prepared by subjecting sterile media to ultracentrifugation. **b** EVs were isolated from bacterial cultures via ultracentrifugation. The pelleted EVs were reconstituted in 30 μ l PBS and EV yields were determined by nanoparticle tracking analysis (NTA) for each time point as indicated on the x-axis (n=4 biological replicates). In addition, sterile media controls were subjected to the same EV isolation protocol described above and NTA was performed (n=3 independent media preparations, nd: not detected).

Since > 99 % of the particles detected by NTA in our EV preparations, isolated using the ultracentrifugation-based protocol, were below 400 nm, it can be assumed that larger particles, such as flagella, did not constitute a significant component of the NTA counts (Explanatory Fig. 5a, b).

Explanatory Fig. 5. Extended size distribution analysis of EVs. Size distributions of (a) EVs_{LB} and (b) EVs_{MOD} shown over 0-2000 nm demonstrate absence of larger particle structures beyond the EV size range (< 400 nm).

8. It is better to show TEM image of a wider view rather than showing one image of a particle. From the image provided it is hard to tell if they are EV or not. Better images might be obtained if ultrathin sections are not used.

Author response: We appreciate the reviewer's suggestion and performed additional TEM analyses employing the drop-on-grid method. Representative images of EVs_{LB} and EVs_{MOD}, obtained by means of the drop-on-grid method, are presented in **new Supplementary Fig. 2c, d**, together with the corresponding negative controls from the sterile media (**see new Supplementary Fig. 2e, f**).

The respective section in the manuscript now read as follows:

Lines 95-104: In addition, we employed two complementary transmission electron microscopy (TEM) approaches: (i) imaging of ultrathin sections prepared from resin-embedded *B. cereus* EVs that were subsequently stained (Supplementary Fig. 2a, b), and (ii) the drop-on-grid method, in which *B. cereus*

EVs were directly dropped onto TEM grids (Supplementary Fig. 2c, d). The TEM analyses confirmed that *B. cereus* EVs deriving from both media are lipid-bilayer bordered entities that are sphere-shaped. In the sterile media (LB, MOD) controls, no particle-like structures were detected by TEM (Supplementary Fig. 2e, f).